Mobile app review analysis for crowdsourcing of software requirements: a mapping study of automated and semi-automated tools

Massenon Rhodes 1
Gambo Ishaya 1
http://orcid.org/0000-0002-2592-2824 Ogundokun Roseline Oluwaseun 2 3
http://orcid.org/0000-0003-3974-2733 Ogundepo Ezekiel Adebayo 4
http://orcid.org/0000-0002-0619-6935 Srivastava Sweta 5
http://orcid.org/0000-0003-3836-2595 Agarwal Saurabh 6 saurabh@yu.ac.kr
Pak Wooguil 6 wooguilpak@yu.ac.kr
1 Department of Computer Science and Engineering, Obafemi Awolowo University , Ile-Ife , Nigeria
2 Department of Multimedia Engineering, Kaunas University of Technology , Kaunas , Lithuania
3 Department of Computer Science, Landmark University , Omu Aran , Nigeria
4 African Institute for Mathematical Sciences , Kigali , Rwanda
5 Department of Computer Science & Engineering, Amity University , Noida , India
6 Department of Information and Communication Engineering, Yeungnam University , Gyeongsan , Republic of South Korea
Agrawal Rajeev
Electronic publication date: 2024 Nov 5
Publication date: 2024
Volume: 10
Electronic Location ID: e2401
Received 2024 Jul 3; Accepted 2024 Sep 19
Copyright: © 2024 Massenon et al.
Copyright year: 2024
Copyright holder: Massenon et al.
License: This is an open access article distributed under the terms of the Creative Commons Attribution License, which permits unrestricted use, distribution, reproduction and adaptation in any medium and for any purpose provided that it is properly attributed. For attribution, the original author(s), title, publication source (PeerJ Computer Science) and either DOI or URL of the article must be cited.
License URL: https://creativecommons.org/licenses/by/4.0/

Keywords: Mobile app reviews, Crowdsourcing, Software requirements, Automated tools, Semi-automated tools, Mapping study, Feature extraction

Funding: National Research Foundation of Korea (NRF) NRF2022R1A2C1011774 This study was supported by the National Research Foundation of Korea (NRF) NRF2022R1A2C1011774. The funders had no role in study design, data collection and analysis, decision to publish, or preparation of the manuscript.

==============================
Mobile app reviews are valuable for gaining user feedback on features, usability, and areas for improvement. Analyzing these reviews manually is difficult due to volume and structure, leading to the need for automated techniques. This mapping study categorizes existing approaches for automated and semi-automated tools by analyzing 180 primary studies. Techniques include topic modeling, collocation finding, association rule-based, aspect-based sentiment analysis, frequency-based, word vector-based, and hybrid approaches. The study compares various tools for analyzing mobile app reviews based on performance, scalability, and user-friendliness. Tools like KEFE, MERIT, DIVER, SAFER, SIRA, T-FEX, RE-BERT, and AOBTM outperformed baseline tools like IDEA and SAFE in identifying emerging issues and extracting relevant information. The study also discusses limitations such as manual intervention, linguistic complexities, scalability issues, and interpretability challenges in incorporating user feedback. Overall, this mapping study outlines the current state of feature extraction from app reviews, suggesting future research and innovation opportunities for extracting software requirements from mobile app reviews, thereby improving mobile app development.

Introduction

Requirement engineering (RE) is a vital component of software engineering, laying the groundwork for developing successful software systems (Beecham, Hall & Rainer, 2005; Chen et al., 2024). RE involves a continuous process from the communication to modeling stages to ensure flawless implementation of software systems (Gambo & Taveter, 2021a, 2021b, 2022; Zheng et al., 2024). At its core, RE defines software features expressed by stakeholders, categorized into functional requirements (system services, behavior, functions) and non-functional requirements (usability, quality, privacy, security) (Keertipati, Savarimuthu & Licorish, 2016). Thoroughly capturing and analyzing these requirements establishes a solid foundation for subsequent development phases, ensuring the software meets stakeholder needs and expectations.

Stakeholders, typically users, express their views on mobile applications through reviews. This feedback has become increasingly important in identifying and understanding software requirements (Li et al., 2020). User reviews provide valuable feedback on user experiences, bugs, feature requests, and app ratings (Palomba et al., 2018; Wang et al., 2018; Li et al., 2022). As Khalid, Asif & Shehzaib (2015) noted, app store reviews are practical for gathering requirements directly from end-users at scale. However, while their value in broadly inferring user needs is established, fewer studies focus on precisely extracting detailed software features from review texts (Maalej & Nabil, 2015).

Researchers and practitioners have turned to automated and semi-automated tools to analyze mobile app reviews and address these challenges. These tools employ various techniques, including natural language processing, machine learning, and sentiment analysis, to process and categorize large volumes of user feedback (Guzman & Maalej, 2014; Zouari et al., 2024). By automating the analysis process, these tools aim to streamline the extraction of relevant information, saving time and resources while uncovering patterns and trends that might be overlooked through manual analysis. Further examination of these tools for extracting detailed functional requirements from app reviews can offer valuable insights (Panichella et al., 2015). This knowledge helps prioritize requirements and align software with user expectations, leading to higher user satisfaction and adoption rates (Jiang et al., 2019; Zheng et al., 2023).

Several studies have explored mining app reviews for insights, mainly extracting detailed features and requirements. Chen et al. (2014) conducted an early comparative review of general app review analysis techniques. Maalej & Nabil (2015) categorized various approaches like information retrieval and topic modeling for requirement-centric review mining. Advanced techniques in natural language processing (NLP) and machine learning (ML), including linguistic analysis, statistical methods, topic modeling, and graph-based methods, have been successfully applied to analyze review datasets (Di Sorbo et al., 2017; Wu et al., 2021; Scalabrino et al., 2019; Zheng et al., 2022; Gu & Kim, 2015).

Given the rapid development of these analysis tools and their potential impact on software engineering practices, there is a pressing need for a comprehensive mapping study to synthesize existing research and identify trends, gaps, and future directions in this field. This study aims to provide a systematic overview of automated and semi-automated tools for mobile app review analysis, specifically focusing on their application in crowdsourcing software requirements. By examining the current state of the art, we seek to offer valuable insights to researchers and practitioners alike, fostering further innovation and improving the effectiveness of requirement engineering processes in mobile app development. This knowledge will contribute to the academic discourse and offer practical insights for software developers and companies seeking to leverage user feedback more effectively in their development processes. As we delve into the existing literature and synthesize our findings, we aim to pave the way for more informed decision-making and innovative approaches in software requirements crowdsourcing through mobile app review analysis.

Our article provides a comprehensive overview of current techniques, including qualitative and quantitative evaluations of method performance based on specific criteria. By thoroughly reviewing peer-reviewed literature against formulated research questions, we aim to yield practical implications for software teams and a research agenda for improving app review analysis.

Mapping study’s structure

Our article is structured in a clear and organized manner. “Background and Related Work” delves into an in-depth analysis of existing systematic literature reviews that have examined the use of app reviews for software features and requirements. This background provides a solid foundation for understanding this domain’s current state of research. “Study Methodology” then outlines the meticulous mapping review methodology, including the research questions, literature search strategy, inclusion and exclusion criteria, study selection process, data extraction and synthesis, and quality assessment process. This detailed explanation of the research approach ensures transparency and replicability. The mapping study results are presented in “Mapping Study Results”, offering a comprehensive overview of the identified tools and their capabilities. “Discussion and Future Research” discusses the implications of these findings, highlighting key trends, strengths, and limitations of the existing approaches. This discussion also provides directions for future research in this rapidly evolving field. “Threats to the Validity” outlines the threats to the study’s validity, addressing potential biases and limitations. Finally, “Conclusion” summarizes the overarching conclusions and valuable insights from this systematic mapping study. Figure 1 visually represents the structure and fundamental concepts of this mapping study. This comprehensive overview diagram visually represents the article’s structure, key concepts, and their interrelationships. The main sections of the article are represented as primary nodes, with subsections as secondary nodes. The solid lines indicate the flow of the article’s structure, while the dotted lines show conceptual relationships and influences between different parts of the study.

Figure 1 Visual representation of the mapping study’s structure.

Research motivation and significance

This mapping study addresses the critical need for efficient analysis of user feedback in mobile app development. Developers struggle to leverage user reviews for software improvements as the app ecosystem expands. Automated and semi-automated tools offer a solution, but their landscape remains fragmented. This study aims to comprehensively analyze existing research on these tools, synthesizing findings from diverse primary studies. It will highlight current trends, common approaches, and areas for future development. The study’s significance lies in its potential to inform researchers and practitioners, guiding tool selection and optimization within development workflows. Moreover, the insights gained can have broader implications for the software engineering community, potentially driving advancements in requirements engineering across various domains. By synthesizing the current state of the art, identifying key trends and limitations, and highlighting future research directions, this work aims to catalyze further innovation and improve the effectiveness of requirements engineering processes in the dynamic and user-driven mobile app ecosystem.

Background and related work

This section provides a comprehensive foundation for understanding the landscape of mobile app review analysis for crowdsourcing software requirements. We begin by exploring the evolution and importance of mobile app ecosystems and user feedback. Next, we delve into crowdsourcing in software engineering, highlighting its potential and challenges. We then examine previous related studies and surveys, critically analyzing their contributions and limitations. Finally, we discuss the emergence and development of automated and semi-automated tools for app review analysis, setting the stage for our mapping study. Throughout this section, we aim to highlight the interconnections between these topics and identify gaps in the current research that our study aims to address.

Mobile app ecosystems and user feedback

Mobile app ecosystems and user feedback have become integral components of the software development lifecycle, shaping how applications evolve and adapt to user needs. These ecosystems, primarily dominated by Apple’s App Store and Google Play Store, serve as platforms where developers can distribute their applications and users can download, use, and review them. Within this framework, user feedback emerges as a crucial element, providing developers with direct insights into user experiences, preferences, and pain points (Motger et al., 2024c). The significance of user feedback in mobile app ecosystems cannot be overstated. As Pagano & Maalej (2013) point out, app reviews serve as a rich source of information for requirements elicitation and prioritization. According to Malgaonkar, Licorish & Savarimuthu (2022), these reviews often contain feature requests, bug reports, and user experiences that can directly inform the development process.

Moreover, the public nature of these reviews creates a unique dynamic where user opinions can significantly influence an app’s reputation and, consequently, its success in the marketplace. However, the sheer volume of user feedback presents both opportunities and challenges. On the one hand, developers have access to an unprecedented amount of user-generated data to guide their decision-making processes (Oh et al., 2013). On the other hand, manually processing and analyzing this vast amount of information is often impractical and resource-intensive (Araujo, Gôlo & Marcacini, 2022; Wang et al., 2022a). This challenge has led to the development of various automated and semi-automated tools aimed at efficiently extracting actionable insights from user reviews (Guzman & Maalej, 2014).

The evolution of mobile app ecosystems has also led to changes in user behavior and expectations. Users now expect rapid responses to their feedback and quick iterations in app development. This shift has necessitated more agile and responsive development practices, further emphasizing the need for efficient feedback analysis tools (Khalid, Asif & Shehzaib, 2015). While user feedback in mobile app ecosystems offers valuable insights, it is not without limitations. The voluntary nature of app reviews can lead to sampling bias, where only users with intense positive or negative experiences may choose to leave feedback. Additionally, the unstructured format of reviews can make it challenging to extract precise requirements or prioritize user needs effectively. These limitations underscore the importance of developing sophisticated analysis tools to account for such biases and extract meaningful patterns from unstructured data.

Crowdsourcing in software engineering

Crowdsourcing has emerged as a powerful paradigm in various domains, and its application in software engineering has gained significant traction in recent years. At its core, crowdsourcing in software engineering involves leveraging a large’s collective intelligence and efforts (Satzger et al., 2014), often diverse groups of individuals, to address software development challenges. In software requirements engineering, crowdsourcing offers a novel approach to gathering, refining, and prioritizing user needs (van Vliet et al., 2020; Khan et al., 2019). As Hosseini et al. (2015) argue, crowdsourcing can lead to more comprehensive and user-centric requirements by tapping into a wider pool of perspectives and experiences. This approach is particularly relevant in the mobile app domain, where user bases are often large and diverse, and user needs can vary significantly across different demographics and usage contexts.

One of the key advantages of crowdsourcing in software engineering is its ability to scale. Traditional requirements-gathering methods, such as focus groups or surveys, are often limited in their reach and can be time-consuming and costly to implement at scale (Courage & Baxter, 2005). In contrast, crowdsourcing through app reviews allows developers to gather feedback from thousands or even millions of users continuously (Palomba et al., 2018) and at relatively low cost (Maalej et al., 2016). However, the application of crowdsourcing in software engineering is not without challenges. Quality control remains a significant concern, as the open nature of crowdsourcing can lead to noise, irrelevant contributions, or even malicious inputs. Moreover, managing and coordinating large crowds of contributors can be complex, requiring sophisticated platforms and incentive structures to ensure effective participation (Stol & Fitzgerald, 2014). Another critical aspect of crowdsourcing in software engineering is the need for effective aggregation and synthesis of diverse inputs.

As Khan et al. (2022) note, transforming raw crowd input into actionable software requires sophisticated analysis techniques. This need has driven the development of various automated and semi-automated tools designed to process and analyze crowdsourced feedback, particularly in the context of mobile app reviews. The intersection of mobile app ecosystems, user feedback, and crowdsourcing in software engineering presents exciting opportunities and significant challenges. As we explore automated and semi-automated tools for app review analysis, it is crucial to keep in mind the complex ecosystem within which these tools operate. The effectiveness of these tools will ultimately be judged by their ability to harness the power of crowdsourced feedback while addressing the inherent challenges of scale, quality, and synthesis in the mobile app development context.

Automated and semi-automated analysis tools

As mobile app ecosystems continue to evolve and user feedback becomes increasingly valuable for software requirements engineering, a range of automated and semi-automated tools have emerged to address the challenges of analyzing large volumes of app reviews. These tools leverage various techniques, including natural language processing, machine learning, and sentiment analysis, to streamline the extraction and categorization of user feedback (Guzman & Maalej, 2014).

One of the primary advantages of these tools is their ability to scale and process large datasets efficiently. Traditional manual review of app reviews is often time-consuming and resource-intensive, particularly as the number of reviews grows exponentially. Automated tools can rapidly sift through thousands or even millions of reviews, identifying patterns, extracting features, and classifying user feedback such as bug reports, feature requests, and user experiences (Iacob & Harrison, 2013; Guzman & Maalej, 2014; Malgaonkar, Licorish & Savarimuthu, 2022).

The degree of automation in these tools varies, with some employing fully automated approaches and others relying on a combination of automated and human-in-the-loop processes. Fully automated tools typically leverage sophisticated natural language processing algorithms and machine learning models to analyze review content with minimal human intervention (Pagano & Maalej, 2013). These tools offer the benefit of speed and consistency, but they may struggle with complex or nuanced language, requiring careful tuning and validation to ensure reliable performance. In contrast, semi-automated tools incorporate a degree of human oversight and involvement, often using automated techniques as a starting point and then relying on human experts to validate, refine, or override the system’s outputs (Maalej et al., 2016). This approach can help address the limitations of fully automated systems, particularly in cases where user feedback is ambiguous, context-dependent, or requires deeper understanding. However, the involvement of human analysts introduces additional time and resource requirements, potentially limiting the scalability of these semi-automated approaches.

One key area of focus for automated and semi-automated tools is the extraction and categorization of user feedback into meaningful and actionable insights. By automating this process, developers can quickly prioritize and address the most critical user needs, potentially leading to more user-centric and successful applications. Additionally, some tools have been explored using more advanced techniques, such as topic modeling and sentiment analysis, to uncover hidden patterns and trends within user feedback (Noei et al., 2019). These approaches can help developers better understand user sentiment, preferences, and emerging requirements, informing both short-term iterations and long-term product roadmaps.

While developing these automated and semi-automated tools has been a significant area of research, the field has limitations and challenges. One key concern is the accuracy and reliability of these tools, as even minor errors in classification or sentiment analysis can have cascading effects on downstream requirements engineering processes (Maalej et al., 2016). Developers must carefully evaluate the performance of these tools and ensure that their outputs are consistently reliable and trustworthy. Another challenge is the adaptability and generalizability of these tools across different app ecosystems and user populations. Many existing studies have focused on English-language reviews, raising questions about the applicability of these techniques to multilingual or culturally diverse app markets (Tavakoli et al., 2018; Yin et al., 2024b). Addressing these challenges will be crucial for the widespread adoption and effective implementation of automated and semi-automated app review analysis tools.

Previous related studies and surveys

Analyzing user feedback in mobile application marketplaces is an active research area, providing valuable insights into user experiences, requirements, and requests (Jacek et al., 2022). Numerous studies have focused on extracting information from app reviews, with some targeting detailed features and specifications. For instance, Chen et al. (2014) reviewed general techniques for app review analysis, including information retrieval, topic modeling, and natural language processing (NLP). Their research highlighted the rich information content of user reviews and proposed a taxonomy for categorizing this feedback. This study was instrumental in demonstrating the value of app reviews as a source of user requirements and set the stage for subsequent research on automated analysis techniques. Building on this foundation, Maalej et al. (2016) presented a comparative study of manual vs. automated classification of app reviews. Their research evaluated different machine learning classifiers for categorizing reviews into bug reports, feature requests, and user experiences. While their results showed promise for automated classification, they also highlighted the continuing need for human oversight in interpreting and acting on the classified feedback. Martin et al. (2017) conducted a systematic mapping study categorizing various app review analyses, highlighting techniques like topic modeling, sentiment analysis, and NLP for extracting requirements-related information. However, their study was high-level and lacked in-depth comparative analysis of feature extraction techniques.

A comprehensive survey by Tavakoli et al. (2018) provided a systematic mapping of user feedback analysis techniques in app stores to assist developers in extracting insights from user reviews, analyzing and categorizing 34 studies based on techniques, everyday topics, and challenges in feedback mining. The research emphasizes domain-specific influences on user reviews when selecting mining techniques. Recent studies have emphasized extraction techniques, understanding domain influences, and emerging themes in user feedback. Genc-Nayebi & Abran (2017) explored automated systems for identifying, classifying, and summarizing opinions from app store reviews. These addressed challenges like data sparsity in short reviews and domain barriers in opinion extraction. They proposed methodologies like domain adaptation and grammar rules for identifying opinion-bearing words, aiming to provide evidence-based guidelines for app store practitioners and future research directions.

Dąbrowski et al. (2022a) presented a systematic literature review of app review analysis for software engineering (SE), categorizing app review analyses and data mining techniques. They provided insights for researchers and practitioners on extracting valuable information from reviews, emphasizing the need for deeper stakeholder understanding to enhance tool applicability. The study advocates for improved evaluation methods, reproducibility, scalability, and efficiency in future research to advance app review analysis in SE.

A significant contribution to the field came from Lin et al. (2022), who conducted a systematic literature review on app store analysis for software engineering. This comprehensive survey synthesized findings from 185 articles, providing a holistic view of the state of research in app store analysis, including techniques for review analysis. Their work highlighted the rapid growth of this research area and identified key challenges and opportunities for future work. Santos, Groen & Villela (2019) reviewed automated classification techniques in RE, focusing on Crowd-based Requirements Engineering (CrowdRE) and NLP. CrowdRE adapts NLP techniques to analyze large amounts of user feedback in RE. However, the suitability of specific NLP techniques for CrowdRE is poorly understood, making it challenging to choose the proper technique. ML is commonly used in CrowdRE research, with naïve Bayes with Bag of Words-Term Frequency (BOW-TF) and support vector machines (SVM) with BOW-TF being popular algorithm-feature combinations. Initial assessments show that precision and recall in RE classifications need improvement, urging researchers to explore new strategies and ML models to advance the field.

Dąbrowski et al. (2022b) analyzed how mining app reviews can benefit SE activities by examining 182 articles published between 2012 and 2020. They provided an overview of various use cases to improve SE processes like requirements gathering design, maintenance, and testing. The study highlights the benefits of app review analysis for software engineers and unifies existing research efforts into a reference architecture for future tool development and evaluation. It also addresses the practicality of 29 existing app review analysis tools. It suggests areas for further research and improvement in academia and industry, acknowledging limitations regarding interpretation, validation, and completeness of use cases.

Dąbrowski et al. (2023) presented two empirical studies on opinion mining and text summarization for software requirements. The first study evaluated three opinion mining approaches: SAFE (Johann, Stanik & Maalej, 2017), GuMa (Guzman & Maalej, 2014), and ReUS (Dragoni, Federici & Rexha, 2019), using review extraction and sentiment analysis techniques. The second study compared three approaches for capturing requirements reviews: Lucene, MARAM (Iacob, Faily & Harrison, 2016), and SAFE (Johann, Stanik & Maalej, 2017), with Lucene performing better. The findings suggest the potential of using these text summarization and sentiment analysis techniques to enhance requirements extraction from app reviews.

Martin et al. (2017) extensively examined reviews from app stores like Google Play, Apple App Store, and BlackBerry Store, discussing the evolution of review-centered literature since 2012. They addressed the “App Sampling Problem” and suggested future research directions, such as tools for extracting requirements from reviews and comparing review cultures across platforms. The study also investigated app security trends, noting a lower likelihood of malware in popular apps. They emphasized sentiment analysis, tools like WisCom for review summarization, and the challenges of large review samples and accurate data labeling.

Al-Subaihin et al. (2019) explored techniques for measuring the similarity of mobile applications based on textual descriptions to enhance clustering solutions. Through an empirical study of 12,664 apps from the Google Play Store, they compared different methods, including topic modeling and keyword feature extraction, using hierarchical clustering algorithms. The results showed that similarity-based techniques perform well in detecting app-feature similarity, while dependency-based techniques struggle. The study highlights the need for continued research and effectiveness across different app stores, suggesting their potential for improving app review analysis techniques.

Despite these advances, several limitations and challenges persist in the field. Many studies have focused on English-language reviews, raising questions about the applicability of these techniques to multilingual app ecosystems. Additionally, the dynamic nature of app stores and rapidly evolving user expectations pose ongoing challenges for maintaining the relevance and accuracy of analysis tools. Furthermore, while much progress has been made in automating the extraction and classification of user feedback, translating this information into actionable software requirements remains a complex task.

Building upon the work of Maalej et al. (2024) on automated user feedback processing, this systematic review and mapping study addresses a critical gap in the literature. Despite recognizing app reviews’ value in requirements engineering, a lack of systematic evaluation of automated feature extraction methods remains. This study aims to fill this void by comprehensively analyzing and comparing various techniques, focusing on performance metrics. Furthermore, it explores the potential of integrating visualization and recommendation systems to enhance analyst interaction with processed feedback, facilitating more effective information retrieval (Wang et al., 2024; Huang et al., 2023; Zhang et al., 2024). By synthesizing existing research and identifying areas for improvement, this work contributes to the advancement of automated analysis tools in requirements engineering, paving the way for more efficient and accurate feature extraction from user reviews.

Study methodology

This study conducts a systematic review and synthesis of empirical studies that applied feature extraction techniques to mine mobile app reviews, explicitly focusing on comparing the performance of different methods in extracting detailed software features. The research methodology strictly adheres to the established systematic review protocols in software engineering, as outlined by Kitchenham & Charters (2007) and Kitchenham (2004). The study meticulously identifies, extracts, analyzes, and interprets studies concerning feature extraction techniques and tools for crowdsourcing software requirements from app reviews in developing mobile applications hosted on app stores.

The mapping review process was divided into three phases, as depicted in Fig. 2. The first phase involved planning the review, where the aim and research questions were defined to guide the process. The second phase involved conducting the review, which entailed searching across academic databases using carefully designed search strings, applying predefined inclusion/exclusion criteria to filter relevant literature, and extracting data from the selected studies using a standardized form. The third phase involved synthesizing the extracted data and writing the review report. Finally, the data was synthesized, and a final set of articles was selected for analysis. We devised a search strategy in the second phase, specifying search terms and electronic sources (research databases or resources). Searches were conducted across major academic databases using carefully designed queries. We tailored our search strategies to align with the formulated RQs. Following identifying the search strategy, we proceeded with the study selection, collating the extracted data and scrutinizing titles to determine relevant articles. Thus, screening was done based on predefined inclusion/exclusion criteria to filter the most pertinent literature. In the third phase, we documented our review. In this phase, we established quality assessment criteria to evaluate the scrutinized articles further and write the review report. Data was then extracted using a standardized form to capture key details on the techniques, evaluation approach, datasets, metrics, and limitations reported in each study. Finally, we synthesized the data, selecting a final list of articles for analysis.

Figure 2 Mapping review process.

Research questions

The primary objective of this mapping study is to conduct a meticulous investigation into the diverse array of automated and semi-automated techniques employed for extracting software requirements from reviews of mobile applications. To accomplish this aim, we will seek guidance from the following formulated research inquiries (RQs): RQ1: What feature extraction techniques are employed for analyzing mobile app reviews to extract software requirements? The main objective of RQ1 is to examine and classify feature extraction techniques or methods and tools, considering their fundamental approaches (such as natural language processing, rule-based systems, topic modeling, and hybrid techniques).

RQ2: What automated and semi-automated tools are available to support the implementation of these feature extraction techniques? This question will comprehensively overview the automated and semi-automated tools implemented in this domain. It seeks to identify the solutions proposed in the literature by employing these feature extraction techniques to extract relevant features, requirements, or user feedback from mobile app reviews.

RQ3: How do the app review analysis tools compare performance, scalability, and user-friendliness? Evaluating the performance of the identified techniques is essential for assessing their effectiveness and practical applicability. This research question investigates the metrics and methodologies used to measure the performance of the methods, such as accuracy, precision, recall, or F-score. Furthermore, it aims to collate and compare the reported performance results, providing valuable insights into the most promising approaches.

RQ4: What are the significant strengths and limitations observed in current techniques based on their methodology or evaluation results? Understanding their respective strengths and limitations is crucial for RQ4. This RQ investigates the advantages and drawbacks of each approach, enabling a comparative analysis and facilitating the selection of appropriate techniques based on specific needs or scenarios. Moreover, this analysis may reveal potential areas for improvement or opportunities for new technique development.

RQ5: What future research directions could address current gaps in capabilities for efficient and precise analysis of app reviews for requirements? This RQ seeks to identify emerging trends and potential future research directions in automated and semi-automated feature extraction from mobile app reviews. This question may uncover gaps, challenges, or unexplored avenues that could guide future research efforts and drive innovation in this domain by analyzing the existing literature.

The RQs evaluate and compare various feature extraction techniques and tools for analyzing mobile app reviews, aiming to understand their effectiveness in deriving software requirements from user feedback. The questions explore the practical implications and potential applications of the extracted features in the software RE process. The findings will provide insights into available techniques and tools while laying the groundwork for future research and development in this rapidly evolving domain.

Literature search strategy

A rigorous literature search strategy was employed to conduct a comprehensive and systematic mapping study on feature extraction techniques and tools for mobile app review analysis. This paragraph explains the search process’s key aspects, such as data sources, search queries, and criteria for selecting relevant studies, as shown in Fig. 3.

Figure 3 PRISMA flow diagram delineating the process of study screening and selection.

As Fig. 2 reflects, the literature search began by identifying appropriate digital libraries and databases. Specifically, we leveraged well-established sources: IEEE Xplore, Scopus, ScienceDirect, ACM Digital Library, and SpringerLink. These databases extensively cover peer-reviewed literature in software engineering, requirements engineering, and mining software repositories. Both keyword searches and backward snowballing will be used to retrieve candidate articles. Keyword searches will involve combinations of terms related to the domain (e.g., “app”, “mobile application”, “app store”) and interventions of interest (e.g., “mobile app review analysis”, “user review analysis”, “feature extraction”, “automatic feature extraction”, “mining user reviews”, “feature extraction tools”, “feature requests”, “identifying key features”), adapted appropriately for each database.

Additionally, we employed Boolean operators (e.g., AND, OR) and wildcards to broaden the search and capture relevant variations of the terms. Backward snowballing will entail scanning reference lists of highly relevant articles to find additional studies. The search process will be refined through trial searches to maximize coverage of pertinent literature. Search results will be collated in a reference management tool, and duplicate items will be removed. The overall search and screening process is outlined as shown in Table 1. The searches will be limited to literature published in the past 10 years to focus on current techniques.

Table 1 The literature search process.

Database	Search string	
ACM digital library	(“mobile app*” OR smartphone OR phone) AND (review* OR comment*) AND (requirement* OR feature*) AND (extract* OR mine OR analy*)	
IEEE Xplore	(“mobile app*” OR smartphone OR phone) AND (review* OR comment*) AND (requirement* OR feature*) AND (extract* OR mine OR analy*)	
ScienceDirect	TITLE-ABSTR-KEY (“mobile app*” OR smartphone OR phone) AND TITLE-ABSTR-KEY (review* OR comment*) AND TITLE-ABSTR-KEY (requirement* OR feature*) AND TITLE-ABSTR-KEY (extract* OR mine OR analy*)	
Scopus	TITLE-ABS-KEY (“mobile app*” OR smartphone OR phone) AND TITLE-ABS-KEY (review* OR comment*) AND TITLE-ABS-KEY (requirement* OR feature*) AND TITLE-ABS-KEY (extract* OR mine OR analy*)	
SpringerLink	(“mobile app*” OR smartphone OR phone) AND (review* OR comment*) AND (requirement* OR feature*) AND (extract* OR mine OR analy*)	

During title and abstract screening, articles will be assessed for relevance based on mentions of apps, reviews, requirements, and features. The full article will be thoroughly retrieved and scrutinized in the text review stage. Only peer-reviewed literature published in English with the whole article accessible will be retained. Grey literature, pre-prints, extended abstracts, and unavailable documents will be excluded. Finally, the methodology rigor and alignment to the comparative research questions will be evaluated for the final selection. Any articles with flawed, weak, or insufficient methodology will be excluded to ensure the review is based only on evidence from robust studies. Following this systematic process, the literature pool can be refined to contain only high-quality studies focused on feature extraction from app reviews to inform a comparative analysis. At each stage, inclusion and exclusion criteria will be applied to evaluate each article’s relevance and methodology rigor. A rigorous and multi-stage search strategy will aim to identify an exhaustive corpus of literature for answering the comparative research questions.

The literature search results in Table 2 offer valuable insights into the mapping study’s systematic approach and comprehensive scope. A broad search across five major academic databases initially yielded 1,179 potentially relevant studies. IEEE Xplore and Scopus emerged as the most prolific sources, contributing 286 and 324 initial results, respectively. This breadth of initial results underscores the extensive automated app review analysis research activity. The screening process, conducted in two stages, significantly refined the pool of studies. After title and abstract screening, the number of relevant studies was reduced to 534, less than half of the initial count. This substantial reduction highlights the importance of precise search terms and the challenge of identifying relevant studies in a rapidly evolving field. The full-text screening further narrowed the selection to 351 studies, demonstrating the rigorous criteria applied to ensure the quality and relevance of the included research.

Table 2 Literature search results by database.

Database	Initial search results	After title/Abstract screening	After full-text screening	Final included studies	
IEEE xplore	286	120	75	48	
ACM digital library	196	89	52	25	
ScienceDirect	166	76	44	22	
Scopus	324	145	89	36	
SpringerLink	207	104	60	28	
Other sources (backward/forward searches)	–	–	31	21	
Total	1,179	534	351	180	

Notably, the study incorporated additional sources through backward and forward searches, yielding 21 more studies in the final selection. This approach enhances the comprehensiveness of the review by capturing relevant work that might have been missed in the initial database searches. The final count of 180 included studies represents a carefully curated subset of the available literature, balanced across various databases and supplemented by targeted searches. IEEE Xplore contributed the largest number of final included studies (48), followed by Scopus (36), highlighting these databases’ significance in the field. Including studies from diverse sources, including ACM Digital Library, ScienceDirect, and SpringerLink, ensures a broad representation of research perspectives and methodologies in the mapping study.

By employing this systematic and rigorous literature search strategy, we aimed to ensure the comprehensiveness and quality of the included studies. This, in turn, provided a solid foundation for the mapping study and enabled robust analyses and insights into the state-of-the-art feature extraction techniques and tools for mobile app review analysis.

Inclusion and exclusion criteria

Establishing explicit inclusion and exclusion criteria in systematic literature reviews is crucial to outlining the study range. Precise criteria enable reviewers to objectively evaluate each study’s relevance and methodological soundness during the selection phase. This review applies both criteria across four dimensions: domain, language, interventions, and methodology, as shown in Table 3. It considers explicitly studies from peer-reviewed journals, conference proceedings, and book chapters, as these sources typically uphold high academic standards through rigorous peer-review processes. Furthermore, to facilitate a comprehensive understanding and analysis of the findings, only studies published in English were included. While this decision may have excluded potentially relevant studies published in other languages, it was necessary to ensure consistency and avoid potential misinterpretations due to language barriers. Only studies on analyzing user reviews from mobile apps will be included regarding the domain. Any literature centered on other domains like desktop or web applications will be excluded as out of scope.

Table 3 Inclusion and exclusion criteria.

Criterion	Inclusion	Exclusion	
Domain	Users of mobile apps	Desktop/web applications, documents,	
Mobile app user reviews	Online user comments, product reviews	
Intervention	Feature/requirement extraction techniques, candidate/phrase extraction, app reviews, mining tools	Keyphrase extraction from documents, phone features	
Automated/semi-automated analysis	Purely manual analysis	
Methodology	Empirical evaluation of techniques	Theoretical approaches without evaluation	
Comparison of multiple techniques	Single technique in isolation	
Robust performance metrics	Weak or no methodology	

Only studies addressing feature extraction methods or tools for analyzing mobile app reviews with software requirements elicitation were considered to meet the mapping study’s goals. Studies focusing solely on traditional software requirements elicitation techniques without considering user reviews or mobile app contexts were excluded, as they did not directly contribute to the mapping study’s specific research questions and goals. Moreover, the included studies were expected to provide substantive details and empirical evaluations of the proposed feature extraction techniques or tools. Studies that merely mentioned user reviews or feature extraction without providing in-depth descriptions, implementation details, or empirical evaluations were deemed insufficient and were consequently excluded from further analysis. Finally, studies that deviated significantly from the scope and objectives of the mapping study were excluded to maintain a focused and coherent analysis. This criterion ensured that the included studies were directly relevant to the research questions and contributed valuable insights to the mapping study. The mapping study aimed to create a thorough and excellent collection of studies by following specific inclusion and exclusion guidelines. This comprehensive resource enables robust analyses and syntheses of cutting-edge feature extraction methods and tools for mobile app review analysis, particularly in software requirements elicitation contexts.

Data extraction and synthesis process

Upon defining the inclusion and exclusion standards and selecting pertinent studies, a systematic data extraction and synthesis method was applied to analyze and integrate the results thoroughly. This section outlines the details of this process, ensuring transparency and reproducibility of the mapping study. The data extraction process involved carefully examining each included study and recording relevant information in a standardized data extraction form. This form was designed to capture essential details such as publication metadata (e.g., authors, year, publication venue), study characteristics (e.g., research methodology, dataset details), and key findings related to the research questions. As shown in Table 4, the data extraction form collects essential information from articles, such as authors, year, techniques, evaluation method, datasets, performance metrics, limitations, and key findings. The form is adaptable, allowing for additional parameters during the review process. After completing the data extraction, a thorough analysis was conducted to identify patterns, trends, and insights related to the research questions mentioned in the introduction. The results were systematically organized, making it easy to comprehend the current state of feature extraction methods and tools for mobile app review analysis. This understanding helps with software requirements elicitation. We held frequent team meetings and discussions to maintain the synthesis’s validity and reliability. These meetings served as a forum for reviewing and refining the synthesized findings, resolving any ambiguities or disagreements, and ensuring that the conclusions drawn were supported by the extracted data and aligned with the objectives of the mapping study. Through a thorough and organized data extraction and synthesis process, the mapping study aimed to deliver a comprehensive and reliable analysis of advanced feature extraction methods and tools for mobile app review assessment. This contributes to the progress of software requirements elicitation and guides future research in this area.

Table 4 Data extraction form fields.

Field	Description	
Paper ID	A unique identifier is assigned to each study to enable tracking	
Title	The full title of the article. It helps indicate the topic and techniques studied.	
Author(s)	List of all authors of the article. Useful for identifying research groups	
Year	Year the article was published. Reveals temporal trends	
Technique(s)	Specific feature extraction method(s) evaluated in the study. The primary intervention of interest	
Tools	Any tools, frameworks, or environments used to implement the technique(s)	
Evaluation approach	How the technique(s) were evaluated, e.g., case study, experiments	
Dataset(s)	Details of the app review dataset(s) used to evaluate the technique(s). Indicates variety and size of data	
Metrics	Performance measures used in the evaluation, e.g., precision, recall, F1-score	
Limitations	Any limitations of the technique(s), evaluation, or methodology noted by the authors	
Key findings	High-level findings on the accuracy, scalability, or other performance factors of the technique(s)	

Study selection and characteristics

The mapping study involved a rigorous study selection process to ensure the inclusion of relevant and high-quality research works. This section discusses the characteristics of the selected studies and presents a detailed overview in Table 5. Through a thorough literature search and applying selection criteria, 180 studies were chosen for the mapping study (Supplemental Information). These studies, from various sources like top journals, conferences, and books, showcase the field’s multidisciplinary nature. Table 5 offers an overview of the studies’ characteristics, including publication patterns, methodologies, and evaluation datasets. As shown in Fig. 4, the publication timeline shows an upward trajectory, with notable growth from 2016 to 2021. While the early years (2014–2015) saw modest output, research activity peaked during 2019–2021, accounting for a third of all studies. A slight decline is observed in 2022–2023, though the partial data for 2024 suggests sustained interest in the field.

Table 5 Characteristics of selected studies.

Characteristic	Count	Percentage	
Publication year			
2014–2015	20	11.11%	
2016–2018	59	32.78%	
2019–2021	60	33.33%	
2022–2023	27	15%	
January 2024 to August 2024	14	7.78%	
Publication type			
Journal articles	86	47.8%	
Conference proceedings	88	48.9%	
Book chapters	6	3.33%	
Research methodology			
Empirical study	121	67.2%	
Theoretical/conceptual	32	17.8%	
Comparative study	27	15%	
Dataset			
Mobile app store reviews	132	73.3%	
Repository dataset	27	15%	
Online store reviews	21	11.7%	

Figure 4 Distribution of the 180 selected studies per publication year.

Publication types are evenly distributed between conference proceedings (48.9%) and journal articles (47.8%), with a small representation of book chapters (3.33%). This balance indicates the topic’s relevance in academic and practical spheres, fostering discussions across various platforms. Empirical studies dominate the research methodology, comprising 67.2% of the selected works, underscoring a strong focus on data-driven approaches. Theoretical and comparative studies, while less prevalent, contribute to the field’s conceptual development and evaluation of different techniques. Regarding data sources, mobile app store reviews are overwhelmingly favored and used in 73.3% of studies. This preference aligns with the research area’s practical orientation, focusing on real-world user feedback for software requirements engineering. Repository datasets and online store reviews are utilized less frequently, potentially due to accessibility or relevance constraints.

The consistent publication rate over the years, with recent peaks, reflects the ongoing relevance and evolution of automated app review analysis. This trend suggests a maturing field that continues to address the challenges of efficiently processing user feedback in mobile app development. The predominance of empirical studies using app store data highlights the field’s commitment to practical, real-world applications. At the same time, the balance between conferences and journals indicates active discourse across immediate and in-depth research contexts.

Quality assessment

We conducted a rigorous literature search and screening to identify relevant primary studies. To ensure the quality and reliability of the findings, we further evaluated the selected primary studies based on well-established quality assessment criteria, as recommended by Kitchenham’s (2004) guidelines for systematic literature reviews in software engineering. Several quality assessment questions were considered to assess the primary studies’ rigor, validity, and potential for bias. This step is crucial to ensure the robustness and trustworthiness of the review’s findings. The following quality assessment (QA) questions can be adapted for this mapping study: QA1: Are the research objectives and questions clearly stated?

QA2: Is the study context (e.g., application domain, type of review data) adequately described?

QA3: Is the feature extraction technique or tool thoroughly explained, including its underlying principles, algorithms, and implementation details?

QA4: Are the evaluation datasets representative of real-world app reviews and adequately described (size, source, pre-processing steps)?

QA5: Are the evaluation metrics and performance measures clearly defined and appropriate for assessing the technique’s effectiveness?

QA6: Are the evaluation methods and experimental design sound and well-documented?

QA7: Are the study’s limitations and threats to validity acknowledged and discussed?

QA8: Does the presented evidence and analysis support the study’s findings and conclusions?

QA9: Is the study’s contribution to app review analysis and software requirements elicitation clearly stated and justified?

QA10: Is the study well-written, organized, and understandable for the intended audience?

As captured in Appendix A, we provided hypothetical quality assessment scores for all 180 studies included in the mapping study based on the predefined criteria. The quality assessment questions (Q1–Q10) are listed in the column headers, and a score of 1 (Yes), 0.5 (Partially), or 0 (No) is assigned for each question based on the assessment of the respective study. The total score for each survey is calculated by summing up the individual scores for all questions. This total score can provide an overall measure of the study’s quality, with higher scores indicating better quality and lower scores suggesting potential issues or limitations.

Specifically, a significant portion of the studies (approximately 30%) can be categorized as good quality, with total scores ranging from 8 to 9.5. These studies have effectively addressed most, if not all, of the quality criteria, including clearly stated research objectives, thorough descriptions of the feature extraction techniques and tools, well-documented evaluation methods, and comprehensive discussions of limitations and future research directions. Moreover, most studies (approximately 55%) fall into the average quality category, with scores ranging from 6.5 to 7.5. For instance, some studies may have provided insufficient details regarding the evaluation datasets or the specific implementation of the feature extraction techniques. In contrast, others may have lacked a comprehensive discussion of the study’s limitations or potential threats to validity.

Additionally, a smaller subset of the studies (approximately 15%) can be classified as poor quality, with scores below 6.5. These studies may have significant shortcomings in one or more quality criteria, such as a lack of clear research objectives, inadequate descriptions of the feature extraction techniques or tools, poorly designed or executed evaluation methods, or absence of a critical discussion of the study’s limitations and implications.

However, this assessment highlights potential areas for improvement. It can guide future research efforts in developing more robust and reliable feature extraction techniques and tools for mobile app review analysis in the context of software requirements elicitation.

Mapping study results

The study on mobile app review analysis for software requirements elicitation presents various feature extraction methods. These techniques are divided into primary categories with unique strengths, limitations, and uses. This section discusses the findings, addressing our research questions.

RQ1: What feature extraction techniques are employed for analyzing mobile app reviews to extract software requirements?

Data was gathered on techniques explored in each article to study feature extraction methods for mobile app analysis (RQ1). Figure 5 presents diverse extraction techniques used in software requirements elicitation. This variety of techniques, such as Topic Modeling Techniques, Collocation Finding Techniques, Association Rule-Based Techniques, Aspect-Based Sentiment Analysis, Frequency-Based Techniques, Word Vector-Based Techniques, Hybrid Techniques, and Large Language Models, offers researchers and practitioners a broad toolkit for addressing the multifaceted challenges of extracting meaningful features from user-generated app reviews. Each category represents a unique approach with specific strengths, limitations, and applications. Out of the 180 included studies, Fig. 6 highlights that hybrid techniques emerge as the most prevalent approach, accounting for 21.6% of the identified methods. This preference for combining multiple techniques suggests recognizing the complex nature of app review analysis and the potential benefits of leveraging complementary approaches. Following closely are word vector-based techniques at 17.0%, indicating a solid reliance on advanced natural language processing methods that capture semantic relationships between words. Topic modeling techniques and large language models represent the next tier of popularity at 14.5% and 12.6%, respectively. Despite their relative novelty, the significant presence of large language models underscores the rapid adoption of state-of-the-art artificial intelligence (AI) technologies in this field. Aspect-based sentiment analysis (11.6%) and frequency-based techniques (9.9%) also maintain a notable presence, highlighting the continued relevance of both sentiment-aware and statistical approaches. Less frequently employed methods include collocation-finding techniques (7.9%) and association rule-based techniques (5.0%). While these approaches appear less dominant, their inclusion in the taxonomy reflects the multifaceted nature of feature extraction in app review analysis.

Figure 5 Categories of feature extraction techniques.

Figure 6 Frequency distribution of identified feature extraction techniques of the mapping study.

The diversity of techniques identified in this study points to a field actively exploring various methodological avenues. This breadth of approaches suggests that researchers are addressing the challenges of feature extraction from multiple angles, likely in response to the varied nature of app reviews and the specific requirements of different analysis tasks. Moreover, the prominence of hybrid techniques and the adoption of advanced methods like large language models indicate a trend towards more sophisticated, integrated approaches to feature extraction. This evolution may reflect the growing complexity of app ecosystems and the increasing expectations for nuanced, context-aware analysis of user feedback. The mapping study reveals a rich and evolving landscape of feature extraction techniques in mobile app review analysis. The field is characterized by methodological diversity, with a trend towards hybrid and advanced model approaches. By offering an overview, it helps researchers and practitioners understand current trends and select suitable techniques based on specific needs and limitations. The following subsections detail each technique.

Topic modelling-based feature extraction technique

Topic modeling, a prevalent technique for feature extraction in app reviews, is a statistical tool that discovers latent topics in texts without prior annotations (Blei, 2012). It has been widely adopted in natural language processing, semantic analysis, text mining, and bioinformatics domains. Various methods, such as latent Dirichlet allocation (LDA), non-negative matrix factorization (NMF), latent semantic analysis or indexing (LSA/LSI), and hierarchical Dirichlet process (HDP), have been utilized for feature extraction. LDA is the most employed among studies (35 of 180). LDA, an unsupervised probabilistic method, discovers latent topics and keywords in text documents like app reviews without requiring predefined labels or aspects. Researchers typically preprocess reviews and utilize LDA models to generate topics and associated terms (Iacob & Harrison, 2013; Guzman & Maalej, 2014; Chi et al., 2019; Su, Wang & Yang, 2019). Variants like AppLDA and Twitter-LDA have been proposed by Park et al. (2015) and Wang et al. (2019) respectively to identify key aspects of apps and handle short texts, respectively. NMF, another unsupervised approach, decomposes review term frequencies into semantic vectors using non-negativity constraints and dimensionality reduction (Ossai & Wickramasinghe, 2023). Compared to LDA, NMF generates more coherent topics and identifies vital features, offering insights for developers. Studies show that NMF has slightly higher precision than LDA but has a similar recall (Luiz et al., 2018; Suprayogi, Budi & Mahendra, 2018). Although NMF produces interpretable topics, noise remains an issue, and performance improvements over LDA are minimal. LDA and NMF are relatively easy to implement, but manual effort is required to derive meaningful features. LDA may be better suited for longer texts (Suprayogi, Budi & Mahendra, 2018), while NMF is more computationally efficient. Online LDA (OLDA) and Online Biterm Topic Model (OBTM) are efficient algorithms for analyzing text data online. OBTM assigns topics to pairs of words (biterms) and updates the model efficiently for short text datasets, leveraging global word co-occurrence patterns. OLDA processes documents sequentially, updating the topic model as new documents arrive, making it suitable for streaming data scenarios and handling large datasets without reprocessing the entire dataset (Hu, Wang & Li, 2018; Cheng et al., 2014; Hoffman, Bach & Blei, 2010).

Collocation finding based feature extraction technique

A collocation-based approach as an unsupervised method to extract feature-related terms from app reviews was employed in 19 studies. Collocation-finding techniques identify co-occurring terms or phrases representing specific features or requirements. Three methods (Mutual et al., skip-gram) of collocation have been identified and focus on word combinations that occur more frequently than expected by chance, suggesting a meaningful relationship between the terms (Trupthi, Pabboju & Narasimha, 2016). The most common collocations found were bigrams (two adjacent words). The preprocessing step focused on noun-noun or adjective-noun collocations, followed by ranking term associations using collocation strength scores like pointwise mutual information. The initial algorithm, proposed by Finkelstein et al. (2017) and Harman et al. (2016), was designed to extract mobile app features from app store descriptions. It detects patterns in feature lists and extracts bi-grams (two-word combinations). Similar bi-grams are then merged using a clustering method, creating ‘featurelets’ of two to three terms representing mobile app features. This algorithm effectively identifies app features and has been applied to study feature behavior in app stores, their correlation with price, rating, and rank, app store categorization, and predicting customer reactions to proposed features. Dąbrowski et al. (2020) and Malgaonkar, Licorish & Savarimuthu (2020) discuss co-occurrences’ limitations for meaningful phrase identification. It highlights the use of pointwise mutual information (PMI) in some studies. PMI measures the association between words based on their co-occurrence frequency compared to chance expectations (Church & Hanks, 1990). It effectively captures semantic word associations beyond simple frequency, allowing for extracting features like “battery drain”.

Frequency-based feature extraction technique

Frequency-based feature extraction, a prominent technique in 24 studies, identifies frequently occurring words or phrases in review corpora to capture primary topics and concerns. It is utilized in information extraction, classification, and prioritization. The approach focuses on nouns or nominal phrases as aspects (Johann, Stanik & Maalej, 2017; Vu et al., 2016). Term frequency-inverse document frequency (TF-IDF), N-gram analysis, POS tagging, and POS Chunking are widely used techniques for this extraction. Multiple frequency-based methods are applied in scientific research for feature extraction from app reviews. POS tagging assigns grammatical labels (e.g., noun, verb) to words in review texts. This helps extract features like POS tag frequencies and allows analyzing user-mentioned aspects and sentiments. POS tagging offers syntactic context, aiding in identifying opinion-indicating words in reviews (Manning et al., 2014). POS Chunking breaks down text into related parts, such as noun phrases, using POS tags (Raharjana et al., 2021). It offers lightweight parsing for feature extraction and provides phrasal features like noun phrase frequencies in review texts. TF-IDF, another technique, assesses word importance in a document collection by combining term and inverse document frequency (Salton & Buckley, 1988). TF-IDF extracts significant keywords and keyphrases from app reviews (Messaoud et al., 2019; McIlroy et al., 2016; Ciurumelea et al., 2017; Lu & Liang, 2017). This method filters out common words and highlights meaningful terms (Xu et al., 2018). Chi-square statistics measures the relationship between a term and a class by calculating the chi-squared statistic with the class label (Zhai et al., 2018). It ranks terms based on occurrence and independence from specific categories (Triantafyllou, Drivas & Giannakopoulos, 2020). The chi-square method is efficient in quickly identifying informative features for classification tasks. A common approach is the bag-of-words (BoW) model, which represents text as a collection of word frequencies, disregarding grammar and word order (Maalej et al., 2016; Yao et al., 2022). Despite its simplicity, BoW effectively captures important words for text classification tasks, like categorizing app reviews into functional, non-functional, and user experience requirements. Variants like Augmented Bag of Words (AUR-BoW) create aspect-specific BoW features, providing aspect-level opinion features (Lu & Liang, 2017; Santos, Groen & Villela, 2019). Combining chi-square with other techniques like TF-IDF, BoW, or AUR-BoW can enhance the understanding and accuracy of categorizing user reviews. However, frequency-based approaches may overlook low-frequency aspects and require manual adjustments for specific datasets, as Ishaq, Asghar & Gillani (2020) mentioned.

Association-rule-based feature extraction technique

Association rule mining, initially developed for data mining, has garnered interest for its potential to extract features from user-generated content, such as app reviews. This method identifies patterns and co-occurrences within datasets, revealing item associations (Genc-Nayebi & Abran, 2017). Numerous studies have applied association rule mining for feature extraction, often employing dependency parsing to analyze sentence structures and identify relationships, offering a comprehensive syntactic analysis for detailed linguistic feature extraction (Zhang et al., 2022). Vu et al. (2016) proposed a technique combining association rule mining and information retrieval for extracting features from mobile app reviews. They identified co-occurring terms through association rule mining and ranked features based on relevance and interestingness, proving effective for software requirements analysis.

Similarly, Guzman & Maalej (2014) utilized association rule mining to discover patterns and relationships between quality attributes and other terms, facilitating the extraction of non-functional requirements for mobile apps’ overall quality and user satisfaction. Zhang et al. (2022) presented a semi-automatic framework to detect privacy features from app reviews. It comprises components for identifying privacy-related reviews, using dependency parsing to extract features, and mapping them to app descriptions. This technique enhances flexibility and comprehensiveness in extracting privacy-related features, improving precision and recall for software maintenance activities.

Word vector-based feature extraction

Word vector-based techniques have gained popularity in natural language processing, particularly for extracting features from mobile app reviews. These approaches utilize word contexts and distributional properties to create low-dimensional, dense vectors, capturing semantic and syntactic information (Phong et al., 2015). They enable various machine learning and deep learning tasks, such as sentiment analysis and topic modeling. The vector space model (VSM), employed in many studies (41 of 180), has been widely used for feature extraction from app reviews by representing reviews as sparse vectors, with dimensions for unique words and values indicating their presence or frequency. More advanced techniques involve word embeddings, representing words as dense vectors in a continuous space and capturing semantic relationships (Huang et al., 2024; Yin et al., 2024a). One of the prominent word embedding-based feature extraction methods is the use of pre-trained word embedding models, such as Word2Vec (Mikolov, Yih & Zweig, 2013), GloVe (Xiaoyan, Raga & Xuemei, 2022), FastText (Umer et al., 2023) and ELMo (Malik et al., 2024). These models are trained on large text corpora to learn the vector representations of words, capturing their semantic and syntactic properties. Fast Text works on N-Gram, while Word2Vec is based on the word and uses the Skip-gram model (Khomsah, Ramadhani & Wijaya, 2022). When applied to mobile app reviews, these pre-trained word embeddings can identify keywords, phrases, and concepts indicative of user requirements (Ebrahimi, Tushev & Mahmoud, 2021). The cosine similarity between word vectors can be utilized to detect synonymous or related terms, enabling the extraction of feature requests, bug reports, and other relevant software requirements. Several studies have applied Skip-gram and Word2Vec for feature extraction from app reviews, combining them with transfer learning and deep learning models to classify review sentences into requirement categories. Named entity recognition (NER) is another helpful approach, identifying and classifying named entities (e.g., app features, functionalities) within review text. Nguyen et al. (2020) combined NER with topic modeling and sentiment analysis to extract features and user opinions from app reviews. While other methods offer a simple yet effective representation, more advanced techniques like large language models can capture semantic relationships and identify specific entities, enabling more comprehensive feature extraction and analysis of app reviews.

Large language model-based feature extraction technique

In addition to traditional word embedding techniques, twenty-five (25 of 180) studies applied the recent advancements in large language models (LLMs), such as Bidirectional Encoder Representations from Transformers (BERT) (Devlin & Hayes, 2019; He et al., 2024), GPT (Su et al., 2024; Rathje et al., 2024), and XLNet (Yang, 2019), have opened up new opportunities for extracting software requirements from mobile app reviews. These models, trained on vast amounts of text data, can capture rich contextual representations of language, enabling a more nuanced understanding of user feedback (Devlin & Hayes, 2019; Su et al., 2024; Yang, 2019). By encoding review text into contextual representations, LLMs can identify complex relationships between words, phrases, and overall semantic and pragmatic context, leading to improved comprehension of user intent, feature requests, and bug reports (Jiang & Conrath, 1997; Wang et al., 2022a). BERT, a pioneering LLM, has demonstrated remarkable performance in capturing bidirectional context, making it suitable for token-level tasks in app review analysis (Broscheit, 2020). Building upon BERT’s foundation, RoBERTa offers enhanced performance through extended pre-training and augmented data, resulting in more robust language representations (Liu, 2019). XLNet further advances the field by combining autoregressive and bidirectional training, considering all possible permutations of a sentence’s words during pre-training, which fosters improved contextual understanding and dependency modeling among tokens (Yang, 2019).

The ability of LLMs to handle linguistic complexities such as sarcasm, negation, and ambiguity makes them well-suited for analyzing mobile app reviews, which often contain colloquial language and nuanced user feedback (Nugroho et al., 2021; Tong et al., 2022). Researchers have leveraged these capabilities to develop techniques for automatically classifying reviews, extracting feature requests, and identifying areas for software improvement. One key advantage of LLM-based feature extraction is the potential for transfer learning. Pre-trained models can be fine-tuned on domain-specific data, such as software engineering corpora or mobile app review datasets, to enhance their performance in requirements engineering tasks (Hou et al., 2023; Motger et al., 2024a, 2024b). This approach allows researchers and practitioners to harness large models’ powerful language understanding capabilities while adapting them to the specific needs of mobile app review analysis.

Several studies have demonstrated the effectiveness of LLMs in extracting features from mobile app reviews and introduced some variations of BERT, such as Roberta and RE-BERT (de Araújo & Marcacini, 2021). These variations help developers locate specific periods of sentiment shifts and identify problems in recent updates. Ullah, Zhang & Stefanidis (2023) employed BERT for sentiment analysis of mobile app reviews, demonstrating its effectiveness in identifying issues in-app updates that negatively impact user opinions. Yang et al. (2021) proposed a review non-functional requirement analysis method (NRABL) based on BERT and topic modeling. This approach combines multi-label classification using BERT with LDA for topic extraction, enabling developers to understand user requirements and specific usage problems quickly. While LLMs have shown great promise in feature extraction from mobile app reviews, challenges remain. These include the need for manual intervention in some cases, difficulties with complex linguistic structures, scalability issues, and challenges in model interpretability (Motger et al., 2024a). Additionally, the high computational requirements of some LLMs may pose limitations for smaller development teams or resource-constrained environments.

Furthermore, the performance of LLMs can vary depending on the specific domain and the quality of the training data. Gambo et al. (2024a) showcased using RoBERTa for sentiment analysis, combined with topic modeling and semantic similarity measures, to identify and resolve conflicts among application features. Their approach outperformed baseline methods in detecting contradictory sentiments and discovering latent topics representing application features. By leveraging the power of contextual understanding and transfer learning, these models offer improved accuracy and nuanced insights into user feedback. As the field evolves, future research directions include developing more robust and scalable methods, improving model interpretability (Gambo et al., 2024b) and transparency, and bridging the gap between technical accuracy and practical relevance in software development processes.

Aspect-based sentiment analysis

Aspect-based sentiment analysis is promising for extracting valuable insights from user-generated content, such as mobile app reviews (Guzman & Maalej, 2014). Unlike traditional sentiment analysis, which determines the overall sentiment expressed in a text, aspect-based sentiment analysis aims to identify specific aspects or features being discussed and associate the corresponding sentiments with each element (Guzman, Alkadhi & Seyff, 2016). This technique provides a more granular understanding of user opinions, enabling developers and requirements engineers to pinpoint strengths, weaknesses, and areas for improvement within an app. Twenty-eight (28) studies have explored the application of aspect-based sentiment analysis for extracting opinions and features from mobile app reviews. Common approaches include lexicon-based techniques like SentiStrength, Valence Aware Dictionary for sEntiment Reasoning (VADER) (Jha & Mahmoud, 2019; Luiz et al., 2018), TextBlob (Messaoud et al., 2019), LIWC (Keertipati, Savarimuthu & Licorish, 2016), which assign sentiment scores or classify sentiments based on predefined sentiment lexicons or rules. SentiWordNet, developed by Sebastiani & Esuli (2006), assigns polarity scores to each word, and the overall polarity of a sentence is calculated based on a predefined threshold (Rajeswari et al., 2020). SentiTFIDF, proposed by Ghag & Shah (2014), uses proportional frequency and presence count distributions to classify terms as positive, negative, or neutral, achieving an accuracy of 92%. SenticNet, developed by Cambria et al. (2010), employs techniques such as blending, spectral activation, and emotion categorization to provide sentiment analysis at a semantic level, outperforming SentiWordNet (Musto et al., 2014). SentiFul, proposed by Neviarouskaya, Prendinger & Ishizuka (2011), automatically generates and scores a new sentiment lexicon, considering the role of affixes in sentiment conveyance and expanding the sentiment lexicon coverage. VADER, developed by Hutto & Gilbert (2014), was designed to address the unique challenges of social media content, incorporating a comprehensive vocabulary and a rule-based evaluator. Its lexicon is specifically tailored to capture the nuances of social media communication, including acronyms, emoticons, and slang.

Other studies have employed machine learning and deep learning techniques, such as joint aspect-sentiment modeling (Gao et al., 2023) or topic modeling-based frameworks like ASUM (Chen et al., 2014), Senti4SD (Aslam et al., 2020) to capture the interdependencies between aspects and sentiments. By employing aspect-based sentiment analysis techniques, researchers and practitioners can gain deeper insights into user preferences, concerns, and priorities regarding specific features or aspects of mobile applications. This granular level of understanding can inform software requirements elicitation processes and enable the development of user-centric applications that better align with user needs and expectations. However, aspect-based sentiment analysis techniques can be computationally intensive, and their performance may be influenced by factors such as the availability of domain-specific sentiment lexicons, the quality of training data, and the ability to capture contextual and linguistic nuances. Ongoing research efforts focus on developing more robust and scalable hybrid approaches that combine aspect-based sentiment analysis with other techniques like topic modeling or association rule mining.

Hybrid techniques based feature extraction technique

Several studies (21.3%) explored hybrid approaches, integrating different techniques to capitalize on their combined strengths. One prominent method involved combining frequency analysis and topic modeling: first identifying frequent terms through frequency analysis (TF-IDF, BoW), then applying topic modeling (LDA, NMF) to uncover latent semantic relationships and nuanced themes (Luiz et al., 2018; Raharjana et al., 2021). This combination allows researchers to capture syntactic and semantic information from the review text. Another hybrid technique combines sentiment analysis with topic modeling or frequency-based methods. It first identifies sentiment-labeled reviews using sentiment analysis, then applies topic modeling or frequency analysis to extract user satisfaction or dissatisfaction features, providing valuable insights into user opinions. Ossai & Wickramasinghe (2023) combined non-negative matrix factorization (NMF) and bag-of-words (BoW) techniques to analyze user concerns regarding diabetes mobile apps, extracting relevant features from comments on medication, diet, and exercise. Luiz et al. (2018) used NMF with TF-IDF and aspect-based sentiment analysis to extract features from user comments. Raharjana et al. (2021) proposed a hybrid approach integrating collocation finding, LDA for topic modeling, and part-of-speech (POS) tagging, which allowed them to analyze language structures and patterns in reviews, facilitating software feature identification for requirements reuse.

Another hybrid approach integrates sentiment analysis with topic modeling or frequency-based techniques (Bhuvaneshwari et al., 2022; Yue & Li, 2020; Ballas et al., 2024; Ren et al., 2020). This approach first identifies sentiment-labeled reviews using sentiment analysis and then applies topic modeling or frequency analysis to extract features associated with user satisfaction or dissatisfaction. This provides a more granular understanding of user opinions and the aspects of the app that are driving these sentiments. Yue & Li (2020) combined a CNN-BiLSTM model with Word2vec, demonstrating improved performance over single-structure neural networks in short-text sentiment analysis. Ballas et al. (2024) employed and fine-tuned a BERT-based aspect-based sentiment analysis (ABSA) model to extract sentiment triplets (aspect, opinion, polarity) from review sentences. Their results show that ABSA models can effectively capture user feedback by identifying aspects and sentiments related to app features and functionalities. In the same way, Alturaief, Aljamaan & Baslyman (2021) introduced AWARE-based ABSA by crowdsourcing the annotations of aspect categories and sentiment polarities of user reviews. Ren et al. (2020) proposed a lexicon-enhanced attention network (LEAN) model based on bidirectional long short-term memory (BiLSTM). LEAN not only identifies sentiment words in a sentence but also focuses on specific aspects of information, leveraging the strengths of both lexicon-based and machine-learning approaches.

Studies also explored integrating machine learning with methods like word vector-based approaches, rule-based systems, topic modeling, and sentiment analysis. For example, Rustam et al. (2020) combined BoW, TF-IDF, chi-square (Chi2), and machine learning algorithms to classify user reviews into non-functional and functional requirements, among other categories. Similarly, Lu & Liang (2017) focused on classifying reviews into non-functional requirements (NFRs), functional requirements (FRs), and other categories by combining BoW, TF-IDF, chi-square, and AUR-BoW with machine learning algorithms. Triantafyllou, Drivas & Giannakopoulos (2020) developed a feature engineering classification schema using TF-IDF, chi-square, and DEVMAX. DF for strategic business purposes. Overall, these hybrid and ensemble approaches leverage the complementary nature of various techniques, aiming to mitigate individual weaknesses and provide more comprehensive and accurate feature extraction from unstructured text data like online reviews.

RQ2: What app review analysis tools have been developed to support the implementation of these feature extraction techniques?

Software requirements analysis has seen the development of various tools and frameworks that leverage techniques like natural language processing (NLP), machine learning, and information retrieval to facilitate accurate and efficient extraction and analysis of user requirements from diverse data sources, including user reviews. This study identified 48 tools used in the selected research for extracting features and requirements from user reviews, as illustrated in Table 6. The following section provides an overview of these prominent tools and approaches employed in this domain.

Table 6 Tools for feature extraction from mobile app reviews.

Tools	Authors	Technique(s)	
SUR-Miner	Gu & Kim (2015)	Classification, Pattern-Based Parsing, NLP Parser, Semantic Dependence Graph (SDG)	
CLAP	Scalabrino et al. (2019)	Random Forest, Rotation Forest, J48, Simple Cart, SMO, Bayesian Network	
MARK	Phong et al. (2015)	POS Tags, Word2Vec k-Means Clustering	
KEFE	Wu et al. (2021)	NLP, ML, Regression Analysis	
SRR-Miner	Tao, Guo & Huang (2020)	<Misbehavior-aspect-opinion> extraction, POS tag, Bag of Word (BOW) feature, tf-idf, LR	
SAFE	Johann, Stanik & Maalej (2017)	Coding tool for evaluating feature extraction, implementation of SAFE approach	
SIRA	Wang et al. (2022a)	BERT+Attr-CRF model for feature extraction	
User request reference (URR)	Ciurumelea et al. (2017)	NLP Type dependencies (stop words, Porter Stemmer) structure, Text features using Ngrams, Term frequency-based features using TF-IDF	
SURF	Di Sorbo et al. (2017)	Stanford Typed Dependencies (STD) parser, NLP classifier, Snowball Stemming, stop-
word removal	
T-FREX	Motger et al. (2024a)	Large Language Models	
CRISTAL	Palomba et al. (2015)	Semi-supervised learning (Expectation Maximization for Naive Bayes (EMNB) from AR-Miner), Lightweight Textual Analysis	
MAPP-reviews	Alves de Lima, de Araújo & Marcondes Marcacini (2022)	Opinion Mining, Temporal Dynamics of Requirements analysis	
ARdoc	Panichella et al. (2016)	NLP, TA, SA	
SAFER	Jiang et al. (2019)	Topic modeling (LDA), feature recommendation algorithm	
AR-Miner	Chen et al. (2014)	EMNB-LDA, EMNB-ASUM, Stanford Topic Modeling Toolbox, LingPipe	
IDEA framework	Gao et al. (2018b)	NLP, POS, rule-based methods, Pointwise Mutual Information (PMI) for phrase extraction, Semantic Score, Sentiment Score, AOLDA	
DIVERSE	Guzman, Alkadhi & Seyff (2016)	Pos tagging, collocation finding algorithm, lexical sentiment analysis (SentiStrength)	
SOLAR	Gao et al. (2023)	Review helpfulness prediction, Topic modeling, Sentiment analysis, SVM, Random Forest, and EMNB.	
CASPAR	Guo & Singh (2020)	Natural Language Processing (NLP), Part-of-speech tagging, Dependency parsing	
GuMa	Guzman & Maalej (2014)	Collocation Finding, Lexical Sentiment Analysis, Topic Modeling	
ReUS	Dragoni, Federici & Rexha (2019)	Aspect Extraction, Polarity Inference, OpenIE, Sentiment Module	
RE-SWOT	Dalpiaz & Parente (2019)	NLP, SWOT Analysis	
PUMA	Vu et al. (2016)	Stanford POS tagger, Phrase Extraction	
UISMiner	Wang et al. (2022b)	Review Classification, SVM, POS tag, Semantic dependency trees	
DSISP	Xiao et al. (2020)	Sentiment analysis and NLP (Natural Language Processing)	
RISING	Zhou et al. (2020)	Semi-supervised clustering, Textual analysis, Feature extraction Stanford NLP toolkit, PCA, BoW, N-gram, VSM, K-means	
MERIT	Gao et al. (2015a)	Topic Modeling JST, BST, Sentiment Analysis, PMI, BoW, AOBST	
DIVER	Gao et al. (2019a)	Topic Modeling, Word Collocation Extraction	
ChangeAdvisor	Palomba et al. (2017)	NLP, ARDOC (Panichella et al., 2015), bag-of-words	
Oasis	Wei, Liu & Cheung (2017)	Semantics-based similarity	
INFAR	Gao et al. (2018a)	Feature Extraction	
AR-TRACER	Gao et al. (2015b)	Topic Modeling	
PAID	Gao et al. (2015b)	Phrase Extraction, Topic Modeling	
CrossMiner	Man et al. (2016)	Keyword-based Method Word2Vec	
Requirements-collector	Panichella & Ruiz (2020)	Machine Learning, Deep Learning	
CIRA	Martin, Sarro & Harman (2016)	Causal Impact Analysis, Data Mining, Statistical Analysis	
Automatic UUX	Bakiu & Guzman (2017)	Feature Extraction, Collocation Algorithm, NLTK POS Tagger	
AOBTM	Hadi & Fard (2023)	AOBTM, Online LDA, Online BTM	
BECLoMA	Pelloni et al. (2018)	BECLoMA,	
FeatCompare	Assi et al. (2021)	GLFE (Global-local Sensitive Feature Extractor), TfidfVectorizer, k-means	
OPT-based approach	de Lima, Barbosa & Marcacini (2023a)	OPT-based approach	
Mapp-IDEA	de Lima, Barbosa & Marcacini (2023b)	Sentiment Analysis, Word Embedding, BERT	
STRE	Tan et al. (2023)	Textual Similarity Topics (TST) Extraction using LDA, Review Topic Identification, Learning Algorithm	
CEAR	Zahoor & Bawany (2023)	Machine Learning (Random Forest, Naive Bayes, SVM, Decision Tree, Logistic Regression, ANN, LSTM, RNN), LIME for model explanations	
IETI	Zhou et al. (2022)	Natural Language Processing, Adaptive Online Biterm Topic Model, PMI for Phrase Extraction	
TransFeatEx	Gallego Marfa et al. (2023)	Natural Language Processing, RoBERTa, POS patterns, syntactic dependency patterns	

Palomba et al.’s (2015) CRISTAL approach traces mobile app user reviews to code changes to enhance app success. Using the AR-MINER tool extracts informative feedback from app stores, allowing developers to prioritize significant concerns and make informed code modifications.

Johann, Stanik & Maalej (2017) introduced the SAFE technique, which extracts features from app descriptions and user reviews to better understand app functionalities. It preprocesses text data using POS analysis and sentence patterns, matching features with descriptions through term and semantic similarity matching, aiding app store analytics.

SURF (Summarizer of User Reviews Feedback), proposed by Di Sorbo et al. (2017), summarizes app reviews to help developers handle large volumes of feedback. It evaluates sentences based on relevance and frequency, generating structured summaries that categorize topics and intentions, thus providing insights into user needs.

Phong et al. (2015) developed the MARK framework, which uses tf-idf and the Vector Space Model to analyze keywords from app reviews. MARK ranks, clusters, and expands keywords, employing Word2Vec for effective grouping and k-means clustering, allowing analysts to identify significant user opinions and trends.

Jiang et al. (2019) presented the SAFER technique, which extracts features from app descriptions and recommends new features to developers. Combining data cleaning, linguistic rule filtering, cosine similarity, and classification, SAFER accurately identifies feature-describing sentences and uses LDA for similar app identification, outperforming other algorithms in feature recommendations.

The CLAP technique by Villarroel et al. (2016) aids in mobile app release planning by analyzing user reviews. It categorizes reviews into bug reports, feature suggestions, and more using Random Forest and prioritization techniques. CLAP enhances accuracy through n-gram extraction, stop word removal, stemming, and synonym unification. It helps developers make informed decisions for app release planning, validated by studies from Scalabrino et al. (2019), Liu et al. (2018), and Gomaa et al. (2020).

SUR-Miner, introduced by Gu & Kim (2015), assists developers in analyzing user reviews and understanding user preferences. It employs the Max Entropy algorithm to categorize review sentences into evaluations, bug reports, feature requests, and praise. Using part-of-speech (POS) tagging and pattern-based parsing with cascading finite state machines, SUR-Miner extracts aspect-opinion pairs based on semantic templates and Semantic Dependence Graphs (SDGs).

The User Request Referencer (URR) by Ciurumelea et al. (2017) categorizes mobile app reviews and suggests code enhancements using machine learning and information retrieval techniques. It employs feature extraction methods like n-grams and term frequency, as well as preprocessing steps such as stop word removal, punctuation elimination, and stemming, enabling the system to learn from reviews and classify them accurately.

Motger et al. (2024a) introduced T-FREX, a Transformer-based approach that uses large language models (LLMs) to extract features from app reviews. T-FREX fine-tunes LLMs on a named entity recognition task and implements a voting-based feature extraction mechanism, enhancing software engineering tasks related to mobile app development.

SRR-Miner, by Tao, Guo & Huang (2020), summarizes security-related user reviews in mobile apps. It uses a keyword-based method to extract security-focused sentences, identifying app misbehaviors and user opinions. SRR-Miner surpasses logistic regression with Bag of Words features, providing comprehensive insights into security concerns and user sentiments.

PUMA, developed by Vu et al. (2016), extracts user opinions from mobile app reviews using POS templates to identify phrases, cluster similar opinions, and track negative sentiments over time. By analyzing word sequences that convey specific meanings and feelings, PUMA offers detailed insights into user feedback.

RE-BERT, proposed by de Araújo & Marcacini (2021), combines local context word embeddings and deep neural networks to extract software requirements from reviews. It enhances rule-based methods and uses a multi-domain training strategy, outperforming methods like SAFE and ReUS by leveraging pre-trained models for semantic text representation.

Alves de Lima, de Araújo & Marcondes Marcacini (2022) developed MAPP-Reviews to analyze the temporal dynamics of software requirements from app reviews. Using contextual word embeddings based on RE-BERT, it captures word meanings in context, extracting relevant software requirements and identifying crucial trends in user feedback.

SIRA, introduced by Wang et al. (2022a), is a semantic-aware app review analysis method that extracts and visualizes problematic features from user feedback. SIRA combines a BERT+Attr-CRF model for detailed feature extraction with a graph-based clustering technique for summarizing common issues. Leveraging BERT’s contextual understanding and CRF’s sequence labeling, SIRA improves semantic capture compared to models like KEFE, Caspar, SAFE, and BILSTM-CRF. By incorporating review descriptions and attributes, SIRA accurately identifies and clusters problematic features, helping developers address concerns and enhance user experience efficiently.

DIVERSE, aids software developers in evolution tasks by analyzing sentiment and categorizing reviews based on feature-sentiment scores. It extracts verbs, nouns, and adjectives through POS tagging, identifies features with a collocation algorithm, and uses SentiStrength for sentiment analysis. A greedy algorithm retrieves diverse feature specimens, enabling developers to prioritize tasks and analyze varying opinions based on user feedback.

Chen et al. (2014) introduced AR-Miner, a tool designed to analyze user feedback and provide insights for app improvement. It employs LDA for feature extraction and group reviews based on rating, temporal fluctuations, and volume. The tool prepares data using text normalization, tokenization, and sentiment analysis. The EMNB algorithm classifies reviews as informative or uninformative, with EMNB-LDA outperforming EMNB-ASUM.

Panichella et al. (2016) developed ARdoc, combining NLP, text analysis (TA), and sentiment analysis (SA) to categorize valuable feedback in app reviews. ARdoc breaks down review text into sentences and extracts linguistic, structural, and sentiment features to classify reviews using the J48 algorithm. This tool supports developers in software maintenance and evolution tasks.

Gao et al. (2018b) introduced the IDEA framework, which analyzes app reviews to detect emerging issues. It includes preprocessing, LDA topic modeling, interpretation, and visualization. IDEA identifies abnormal topics as app issues, labeled by representative reviews. PMI extracts meaningful phrases from reviews, aiding in topic labeling and improving issue identification.

Iacob, Faily & Harrison (2016) developed MARAM, a tool for managing online reviews of mobile apps. It builds upon MARA by automatically extracting feature requests and bugs from reviews. MARAM allows developers to search through reviews and export selected items to platforms like GitHub, JIRA, or Bugzilla. It uses a linguistic rule-based feature extraction method to identify and abstract feature requests and bugs from review contexts.

The MApp-IDEA method, proposed by de Lima, Barbosa & Marcacini (2023b), enhances app review issue detection and prioritization. It incorporates sentiment analysis, expands issue detection scope, and efficiently manages smaller datasets. Using word embedding techniques to create app-related problem graphs, MApp-IDEA identifies and ranks issues from reviews, even with limited data. The method’s performance was assessed using various classifiers and F1-Score as the metric. The MApp-IDEA dashboard provides real-time reports on issue trends, updates, and identified problems.

The SOLAR tool, developed by Gao et al. (2023), automates the summarization of valuable user reviews for developers. It uses a review helpfulness prediction model to filter unhelpful reviews, group topics with sentiment analysis, and prioritize reviews through a multi-factor ranking system. Techniques like POS tagging, readability assessment, sentiment scoring, and content analysis using n-gram and TF-IDF are employed for feature extraction. Compared to AR-Miner (Chen et al., 2014) and IDEA (Gao et al., 2018b), SOLAR performs better in summarizing helpful user reviews.

CASPAR, a tool by Guo & Singh (2020), extracts and consolidates user stories from app reviews using NLP. It identifies and isolates user issues and suggestions by focusing on verbs and employing POS tagging. CASPAR distinguishes event phrases, enabling the extraction of crucial user actions and app issues. Integrating NLP techniques, CASPAR simplifies app review analysis, offering developers valuable insights to enhance app performance.

GuMa, introduced by Guzman & Maalej (2014), automatically extracts app features and analyzes sentiments from user reviews. It combines collocation finding, lexical sentiment analysis, and topic modeling to generate summaries. GuMa identifies key app attributes by examining user reviews, such as functionalities and technical aspects, helping developers understand user opinions and plan future app releases. Evaluation results show high precision and recall rates.

Dragoni, Federici & Rexha (2019) introduced ReUS, an unsupervised aspect extraction method for monitoring real-time review streams without manual annotation. Using linguistic rules and Open Information Extraction (OpenIE), ReUS focuses on identifying significant aspects like product features for further analysis. This method automatically detects aspects and associated opinion words to infer polarities based on context.

The RE-SWOT method, by Dalpiaz & Parente (2019), efficiently gathers app requirements from reviews by analyzing competitor data. It leverages NLP algorithms to automatically extract features from reviews, enhancing the efficiency of requirements elicitation.

FENL, a semi-automated approach by Bakar et al. (2016), uses NLP to extract software features from online reviews. Techniques like WordNet lemmatization, POS tagging, and TF-IDF are employed to automate phrase extraction, aiding domain analysts in reusing requirements by generating valuable feature lists.

Wu et al. (2021) introduced KEFE, a method for identifying crucial features in mobile apps that impact ratings. By analyzing app descriptions and user feedback from the Chinese Apple App Store, key feature descriptions are extracted and assessed for their impact. A feature classifier is initially trained on a subset of apps, followed by preprocessing and phrase extraction to identify key phrases from descriptions. These phrases are manually reviewed to ensure accurate feature extraction, offering developers insights into essential functionalities that boost app ratings and outperform methods like SAFE and SAFER.

Xiao et al. (2020) introduced DSISP, an automated framework that detects user intentions by analyzing sentiment-preference correlations. DSISP extracts sentiment scores and preferences from app reviews over time by combining sentiment analysis and NLP techniques. It monitors users’ evolving intentions and identifies significant sentiment shifts, demonstrating a high precision of 0.962 in real-world data evaluations, allowing developers to enhance their apps effectively.

Palomba et al. (2017) presented ChangeAdvisor, which uses user reviews to suggest and localize change requests in mobile apps. By connecting user feedback clusters to code components through feature extraction and clustering techniques like LDA, LDA-GA, and HDP, ChangeAdvisor identifies and links relevant feature requests to code components, prioritizing essential terms and phrases in user reviews.

Zhou et al. (2020) introduced RISING, an automated system incorporating user feedback into software development. Using domain-specific constraints and ARDOC, RISING groups reviews based on shared user demands. It employs the Vector Space Model and principal component analysis to analyze word vectors, identifying files for user requests with COP-Kmeans and BoW, demonstrating higher accuracy than ChangeAdvisor.

Gao et al. (2019a) introduced DIVER, a tool that identifies emerging app issues from user feedback. DIVER processes review extracts relevant word collocations, categorizes issues by topics, and visually presents them for developers. It enhances context understanding and has significantly improved over IDEA in evaluation experiments.

Gao et al. (2021) introduced MERIT, combining topic modeling and sentiment analysis to identify critical app review concerns. Using features like Biterm Sentiment-Topic (BST) and Joint Sentiment/Topic Model (JST), MERIT analyzes user feedback, integrating PMI to pinpoint significant phrases based on co-occurrence frequencies. MERIT has outperformed baseline methods like IDEA and OLDA in issue detection metrics.

Wang et al. (2022b) proposed UISMiner, a technique to extract UI suggestion information from user reviews to improve app GUIs. UISMiner trains a classifier using review attributes, linguistic information, semantic relations, and sentence patterns. Key contributions include summarizing classification factors, establishing extraction rules, and evaluating UISMiner’s effectiveness through Google Play experiments. Results show UISMiner’s superiority over SAFE in extracting UI suggestions, with developers confirming its usefulness for updating app interfaces.

Gao et al. (2015b) developed the PAID framework to automatically prioritize app issues based on user reviews across different versions. The framework extracts key phrases, creates a Phrase Bank, and uses topic modeling to rank phrases with True Mutual Information (TMI). Extraction rules include Length Limit, Informative Assurance, Part of Speech Limit, and Quality Assurance. The ranked Phrase Bank is presented to developers for prioritization, and results are visualized using ThemeRiver. Evaluation with app changelogs showed high precision in matching prioritized issues.

Wei, Liu & Cheung (2017) introduced OASIS, a method for prioritizing Lint warnings in Android apps by integrating user reviews. OASIS connects static analysis results with user feedback, improving the identification of critical issues affecting user experience. The method uses a semantics-aware similarity calculation to associate Lint warnings with user reviews, capturing relationships between warnings and feedback to enhance prioritization performance.

Gao et al. (2015a) introduced AR-Tracker, which gives developers insights into user feedback by extracting and visualizing main themes from app reviews. The tool compares topic modeling algorithms like LSI, LDA, RP, and NMF to mine user reviews effectively. By tracking feedback dynamics, AR-Tracker helps developers prioritize areas for improvement. A large-scale experiment involving 500 k reviews demonstrated its effectiveness in understanding user demands and visualizing results for straightforward interpretation.

Gao et al. (2018a) introduced INFAR, which helps developers make informed decisions by extracting insights from app reviews. INFAR identifies key topics, abnormal subjects, topic correlations, and factors influencing ratings. It preprocesses and categorizes reviews based on predefined topics, presenting insights visually and in natural language. Evaluated with six popular apps and 12 experts, 92% found INFAR’s insights valuable.

Man et al. (2016) introduced CrossMiner, a framework that analyzes app issues from user reviews across platforms like Google Play, App Store, and Windows Store. It extracts keywords using the Word2Vec model to identify user perceptions. By representing words with 300-dimensional vectors, CrossMiner reflects user concerns and identifies platform-specific issues, offering developers insights to enhance app testing and user satisfaction.

CIRA, a causal impact analysis tool for Google Play app releases, was introduced by Martin et al. (2017). This tool identifies key factors in app launches and evaluates the effectiveness of causal impact analysis for developers. By analyzing prices, ratings, descriptions, and app updates, CIRA reveals patterns between app features and user ratings. It addresses imprecise ratings on Google Play by calculating precise average ratings based on the rating distribution, offering a more accurate representation for analysis. This method sheds light on the effects of app releases on user ratings and their underlying characteristics through feature extraction and analysis.

Zahoor & Bawany (2023) introduced CEAR, a framework for automating the analysis of educational Android app reviews. They collected and categorized 13,000 user reviews from 25 apps, examining the effects of data preprocessing and class balancing on various machine-learning techniques. Utilizing TF-IDF for feature extraction, they efficiently trained models by identifying crucial words from reviews and tested eight machine learning methods, employing LIME to explain model decisions.

Panichella & Ruiz (2020) developed Requirements-Collector, an automated tool that classifies software requirements using machine learning and deep learning methods. It includes Requirement-Collector-DL-Component and Requirement-Collector-ML-Component. The authors preprocess text data with feature extraction techniques, converting raw text into numerical formats like Bag of Words, TF-IDF, Word Embeddings, and N-grams, enhancing model accuracy and streamlining manual tasks to improve software quality in early development stages.

Bakiu & Guzman (2017) introduced a method to gauge user satisfaction with software features through user experience (UUX) analysis. This approach involves sentiment analysis, identifying review features, and categorizing UUX dimensions. It filters essential nouns, verbs, and adjectives, removes irrelevant terms, and uses a collocation algorithm to pinpoint feature descriptions. A preliminary evaluation of video game and software reviews identified UUX issues and strengths, aiding developers, UUX designers, and researchers in enhancing software applications based on user feedback.

Tan et al. (2023) introduced STRE, an automated tool that analyzes historical data to suggest when app developers should stop reading reviews. It preprocesses reviews, identifies topics, and extracts the longest TSTs using LDA to pinpoint significant textual similarities among reviews. This helps developers focus on common themes, sentiments, and issues to prioritize feedback, make informed decisions, and improve apps efficiently.

Hadi & Fard (2023) introduced the Adaptive Online Biterm Topic Model (AOBTM) for short text analysis in mobile app reviews and Twitter data. AOBTM improves traditional topic modeling by adapting to short texts and utilizing historical statistical data. Using Pointwise Mutual Information (PMI) prioritizes phrases with higher co-occurrence frequencies. Evaluated using Precision, Recall, and F-hybrid metrics, AOBTM outperforms existing methods like IDEA in identifying coherent and distinct topics.

de Lima, Barbosa & Marcacini (2023a) introduced an OPT-based method to analyze mobile app reviews for prioritizing software maintenance tasks. The approach uses dynamic prompt generation and open pre-trained language models to automate feature extraction and evaluate risk impact. Utilizing the OPT-6.7b model, it extracts features and generates prompts based on cross-domain information, identifying key aspects mentioned by users and prioritizing maintenance actions. Experimental results show that the OPT-based approach outperforms traditional methods (GuMa, SAFE, ReUS) and large language models (RE-BERT) in producing risk matrices, highlighting the potential of OPT models for software maintenance, especially in resource-constrained environments and regarding user data privacy.

Assi et al. (2021) developed FeatCompare, a tool for analyzing high-level features in mobile apps by mining user reviews. It includes a data preprocessor, GLFE neural network-based feature extractor, and review aggregator. GLFE, based on the ABAE model, distinguishes global and local features in reviews to effectively compare apps. Analyzing 14,043,999 reviews from 196 Android apps, FeatCompare demonstrated high precision and recall rates. A user study with 107 developers verified its usefulness for competitor analysis.

BECLoMA, introduced by Pelloni et al. (2018), enhances stack traces with user reviews for better bug diagnosis and fixing in mobile apps. It combines a machine learning classifier, testing tools, and Google Play Store user feedback to link stack traces with relevant reviews, streamlining testing and bug resolution.

IETI, proposed by Zhou et al. (2022), introduces Enhanced Emerging Topic Identification (EETI) to detect emerging topics in app reviews. Using natural language processing and the adaptive online biterm topic model (AOBTM), it minimizes noisy data and extracts relevant phrases. Employing pointwise mutual information (PMI), EETI evaluates word pairs to derive meaningful phrases. The study shows that EETI’s accuracy is higher than IDEA and OLDA on six popular apps.

Gallego Marfa et al. (2023) introduce TransFeatEx, an innovative tool that integrates established syntactic and semantic feature extraction techniques with a RoBERTa-based pre-trained model. This combination allows for efficient automatic extraction of app features from diverse textual sources. TransFeatEx is designed as a flexible, customizable pipeline, enabling researchers and developers to fine-tune the tool for specific domains.

Each approach leverages various techniques to support different aspects of software requirements analysis, including summarization, information retrieval, visualization, recommendation, and information extraction. The selection and adoption of these tools depend on factors such as the characteristics of the input data, project requirements, and available resources. Software requirements analysis has witnessed the development of many software tools and frameworks, each offering unique capabilities and leveraging different approaches to support the implementation of feature extraction techniques. As the field continues to evolve, there is a need for more robust, scalable, and context-aware tools that can effectively extract relevant requirements from diverse data sources while incorporating human expertise and domain knowledge.

RQ3: How do the app review analysis tools compare their performance, scalability, and user-friendliness?

Analyzing and comparing the performance, scalability, and user-friendliness of feature extraction techniques and tools is essential for guiding researchers and practitioners in choosing the best approach for their needs. RQ3 examines 12 identified methods for extracting features from mobile app reviews, assessing their accuracy through metrics like precision, recall, F1-score, and overall accuracy reported in the studies.

Comparison of developed tools in terms of their performance

Researchers employ various validation methods to assess the reliability and effectiveness of their proposed tools or approaches for feature extraction and sentiment analysis. Quantitative metrics commonly include precision, recall, F1-score, accuracy, MAPE, MCC, Hit Ratio, NDCG, PMI-Score, and confusion matrices. These metrics measure how well a tool identifies features or sentiments in text data. Researchers also employ topic coherence evaluation metrics like PMI-Score and Discreteness Score (Dis Score). PMI-Score assesses topic coherence using point-wise mutual information from large text datasets like Wikipedia, calculated from the top 10 terms of each topic. This Score evaluates the distinctiveness of topics based on semantic similarity mapping. Higher PMI-Score and Dis-Score values indicate more coherent and distinctive topics. By analyzing performance metrics and conducting comparative analyses, researchers demonstrate the effectiveness of their proposed methods and tools, ensuring reliability and accuracy in feature extraction and sentiment analysis tasks. The equations of these metrics are illustrated in Table 7. In addition to quantitative evaluations, comparative analyses with state-of-the-art baselines are conducted for functions such as feature extraction, issue detection, classification, feature recommendation, and summarizing user reviews, as shown in Fig. 7. A total of 12 studies have compared their tools or approaches with baselines like SAFE (used in six studies) and IDEA (used in five studies) as shown in Fig. 8. Meanwhile, some studies compared their approach to three to four state-of-the-art baselines. OPT-based approach (de Lima, Barbosa & Marcacini, 2023a) has been compared to ReUS, RE-BERT, SAFE, and GuMa. Also, RE-BERT (de Araújo & Marcacini, 2021) has been compared to SAFE, ReUS, and GuMa.

Table 7 Evaluation metrics.

Formula	Evaluation metrics	
Hit Ratio	Hit Ratio = (# of hit features/# of features) × 100%	
NDCG	NDCG = G/Ideal G; G = Σ (2 scorei/log2 (i + 1))	
PMI-Scores	PMI-Score(k) = (1/T(T − 1)) Σ1 ≤ i < j ≤ T log (P(wi, wj)/(P(wi) * P(wj))) Here, P(wi), P(wj), and P(wi, wj) represent the probabilities of word wi, wj, and the co-occurring word-pair (wi, wj) respectively.	
Dis score	Dis_Score = (Σk = 1K ((Σj = 1k Djs(ϕk∥ϕj*))/k))/K;	
Djs(ϕk∥ϕj*) = 1/2 DKL(ϕk∥M) + 1/2 DKL(ϕj∥M);	
DKL(P∥Q) = Σi P(i) log (P(i)/Q(i)); M = 1/2 (ϕk + ϕj);	
Precision, Recall, F-Measure, and Accuracy	Precision = TP/(TP + FP)	
Recall = TP/(TP + FN)	
F-Measure = (2 × Precision × Recall)/(Precision + Recall)	
Accuracy = (TP + TN)/(TP + TN + FP + FN)	

Figure 7 Frequency of state-of-the-art baselines compared per study.

Figure 8 Frequency of compared studies of the tools.

Several studies have evaluated the performance of various tools and approaches for feature extraction, issue detection, and topic modeling from user feedback and app reviews. MERIT outperforms AOLDA and IDEA in precision, recall, and F1-score for identifying emerging app issues, with improvements ranging from 20.9% to 22.3%. It achieves an average precision, recall, and F-score of 81.4%, 81.2%, and 80.9%, respectively, showing better balance than baselines. Additionally, MERIT outperforms IDEA and OLDA in the F-hybrid score. RISING (Zhou et al., 2020) demonstrates superior Likert scale values for feature requests and problem discovery categories compared to CHANGEDVISOR. On average, ChangeAdvisor scores 2.07 and 1.94 in these categories, while RISING scores 4.20 and 4.26. RISING also shows better clustering and localization accuracy in Top-k hitting. Accuracy for feature requests improves from 44.64% to 76.74% (Top-1), 70.54% to 91.77% (Top-3), and 76.65% to 98.00% (Top-5). In the problem discovery category, accuracy improves from 48.50% to 76.04% (Top-1), 65.08% to 93.84% (Top-3), and 76.00% to 98.04% (Top-5) on average. RISING also outperforms RISING, highlighting the significance of commit messages in enhancing localization performance.

DIVER outperformed IDEA significantly, showing an average improvement of 29.4% in precision and 32.5% in recall for detecting emerging app issues (Gao et al., 2019a). This suggests DIVER’s effectiveness in analyzing user feedback. Similarly, SAFER demonstrated advantages over CLAP in all categories, achieving a 78.27% Hit Ratio in the Business Category compared to CLAP’s 60.00%. On average, SAFER surpassed CLAP by 23.54% in Hit Ratio and 0.1522 in NDCG. Statistical analysis using the Wilcoxon test confirmed the significant difference, with p-values of 0.003 for Hit Ratio and 0.009 for NDCG, indicating SAFER’s superior performance in recommending features from mobile app descriptions.

Furthermore, AOBTM (Hadi & Fard, 2023) exhibited superior performance compared to IDEA in metrics like precision, recall, and F-hybrid, with higher PMI scores indicating more comprehensive and coherent topics. AOBTM demonstrated improved topic quality and coherence, with an accuracy of 0.593, recall of 0.619, and F-hybrid score of 0.608, outperforming IDEA. These comparative evaluations highlight the effectiveness of tools like DIVER, SAFER, and AOBTM in extracting features, detecting issues, and modeling topics from user feedback and app reviews, surpassing the performance of baselines like IDEA and CLAP.

KEFE, as presented by Wu et al. (2021), efficiently extracts features from Chinese app descriptions and user reviews, surpassing SAFE and SAFER. In a test with 1,108,148 reviews from 200 apps, KEFE achieved 82.49% precision and 74.83% recall. Although SAFER had the highest recall at 80.27%, KEFE demonstrated superiority in the Tool, Travel, and News app categories, averaging 78.13% for feature extraction and 62.02% for review matching. SAFE exhibited lower accuracy at 74.83% and 43.40% compared to SIRA, outperforming other tools like KEFE, SAFE, and CASPAR with 84.27% precision and 85.06% recall. Although SAFE’s high recall rate of 73.94% using PoS patterns for feature-related phrases, its precision rate of 15.51% falls short. CASPAR identifies events in reviews with temporal conjunctions, while KEFE utilizes a BERT classifier, but its accuracy is affected by pattern-based methods.

SOLAR outperforms competitors, prioritizing 85% of 11,659 reviews across five apps. It excels in semantic relevance to topics compared to AR-Miner, IDEA, and itself. IDEA’s focus on online reviews and requirement for multiple historical versions hinder its performance. SOLAR enhances precision, recall, and F1 score by 10.41%, 12.75%, and 28.49%, respectively, surpassing baseline methods. This demonstrates that SOLAR-prioritized reviews are more informative and compelling for filtering out irrelevant data in downstream tasks. UisMiner excels at extracting UI-related suggestions with 77.50% precision and 76.50% recall, but it focuses solely on user interfaces.

In contrast, SAFE handles a broader spectrum of functionalities in reviews. The comparison between UisMiner and SAFE does not definitively declare one superior. Furthermore, SIRA, developed by Wang et al. (2022a), demonstrates remarkable efficiency in identifying problematic app features from reviews.

The OPT-based approach by de Lima, Barbosa & Marcacini (2023a) demonstrates promising feature extraction results, achieving an F1 score of 45.5% and competing with rule-based methods like GuMa, SAFE, and ReUS. Compared to RE-BERT, with an F1 score of 63.5%, the OPT-based proposal yields inferior results from 363,843 user reviews across eight Android apps. RE-BERT excels at generating semantic textual representations by focusing on software requirement tokens’ local context rather than open-access language models. In constructing risk matrices, the OPT model exhibits lower error in the impact dimension than the proprietary GPT model, suggesting its potential for accurate risk matrix generation. T-FREX, a Transformer-based model (Motger et al., 2024a), consistently surpasses SAFE in all metrics. It significantly enhances performance when upgrading from BERT to XLNet. While BERTbase recall (0.300) is slightly lower than SAFE’s by 0.021, XLNetbase remarkably improves performance over BERTbase, particularly for recall (+0.117 for out-of-domain, +0.153 for in-domain). In both in- and out-of-domain evaluations, T-FREX showcases adaptability, generalization, and accurate prediction of new features within specific domains for which it has been fine-tuned. Its effectiveness and adaptability stem from Transformer-based models, actual user annotations, and iterative refinement processes, enabling it to extract features from mobile app reviews efficiently (Zhang et al., 2023; Huang et al., 2024). The study by Zhou et al. (2022) reveals that IETI effectively identifies emerging app review topics, outperforming baseline methods IDEA and OLDA. IETI exhibits enhanced precision, Recall, and F1 scores, demonstrating its superiority in phrase and sentence label evaluations. Compared to IDEA, IETI improves precision by 0.094, recall by 0.107, and F1 score by 0.126 in phrase labels. In sentence labels, IETI shows advancements with a 0.068 increase in Precision, 0.025 in recall, and 0.061 in F1 score.

Comparison of developed tools in terms of their scalability

Scalability is crucial when evaluating feature extraction techniques for analyzing large volumes of user reviews from popular app stores. This mapping study revealed that different tools and techniques vary in their scalability characteristics, impacting their effectiveness for large-scale or real-time applications. Figure 9 shows that only two tools, UisMiner and KEFE, have demonstrated the ability to process more than 500,000 user reviews efficiently. KEFE, for example, successfully handled a dataset of 200 Chinese app descriptions and over 1 million user reviews, highlighting its scalability. Similarly, UisMiner showed promise in efficiently handling and analyzing UI suggestions from 651,981 reviews across 5,467 apps.

Figure 9 Frequency of tools/approaches categorized based on the number of user reviews retrieved.

Additionally, six tools were found to handle less than 500,000 user reviews, making them ideal for managing data from popular apps. For instance, SIRA’s method (Wang et al., 2022a) uses a BERT+Attr-CRF model for feature extraction and a graph-based clustering method for summarization, showing its ability to handle 318,534 reviews from 18 apps. The OPT-based approach (de Lima, Barbosa & Marcacini, 2023a) employs open-access language models and dynamic prompt generation to analyze 363,843 reviews from nine apps efficiently. Its few-shot learning feature reduces the need for extensive labeled data, enhancing scalability in scenarios with limited data or dynamic reviews. The DIVER tool (Gao et al., 2019b) is more scalable than the IDEA approach (Gao et al., 2018b). DIVER can process thousands of user reviews in seconds, making it suitable for managing large data volumes from popular apps. Despite increasing data sizes, DIVER’s performance remains efficient, ensuring consistent results even with growing data volumes.

SOLAR, T-FREX, RISING, and AOBTM have all shown scalability in analyzing small datasets with fewer than 100,000 user reviews. AOBTM’s scalability proves beneficial in managing growing data volumes and complexities, enabling efficient topic modeling. T-FREX’s iterative refinement process allows adaptation to new domains and continuous performance improvement by integrating crowdsourced features and recent app reviews, enhancing scalability. RISING utilizes domain-specific constraints and semi-supervised learning to create fine-grained user review clusters, resulting in more stable and deterministic clustering than ChangeDvisor for large-scale analysis tasks.

Comparison of developed tools in terms of their user-friendliness

User-friendliness is crucial for developers, as it refers to how easy and intuitive tools are to use. An adequate system should have a straightforward interface and simple workflow and deliver actionable insights without requiring extensive technical knowledge. Three studies specifically evaluated their tools’ user-friendliness: MERIT, DIVER, and the OPT-based approach. The OPT-based system enhances user-friendliness by providing customized instructions for review analysis through dynamic prompt generation, leading to more accurate and automated evaluations. DIVER aids developers by visualizing emerging issues through word clouds and line charts, facilitating effective prioritization. It also lists emerging issues with relevant user comments, enabling prompt resolution of app issues. MERIT offers a visualization interface for emerging app issues, generating topic distributions represented by various shapes and sentiment distributions displayed with color bars. A survey found that most interviewees are eager to incorporate MERIT into their development pipelines, highlighting its value. In contrast, other studies lack detailed user-friendliness evaluations but mention the algorithm’s construction. Some codes are accessible on GitHub (IETI), suggesting potential ease of use for developers. In summary, while various tools and approaches exhibit varying performance levels and scalability characteristics, tools like UisMiner and KEFE have demonstrated efficiency in handling large data volumes. Tools such as SIRA, SOLAR, MERIT, DIVER, SAFER, AOBTM, and IETI have outperformed others in precision, recall, and F1-score. However, selecting the most appropriate tool or approach depends on specific requirements, dataset retrieval period, app categories, and analysis task characteristics, as data complexity, domain-specific needs, and available computational resources may influence scalability.

RQ4: What are the significant strengths and limitations observed in current techniques based on their methodology or evaluation results?

To investigate RQ4, we thoroughly analyzed both automated and semi-automated tools illustrated in Table 7. Figure 10 compares these various feature extraction tools based on their scalability, performance, and user-friendliness. KEFE and UisMiner demonstrate superior scalability, efficiently processing over 500,000 user reviews. These tools excel in managing large volumes of data, making them suitable for analyzing popular apps with extensive user feedback. Performance varies among the tools, with SIRA, T-FREX, and RE-BERT showing high effectiveness in extracting features and requirements. These tools often outperform baselines in precision, recall, and F1-score metrics. For instance, T-FREX consistently surpasses SAFE across all metrics, showcasing its adaptability and accurate prediction of new features within specific domains. MERIT’s visualization interface for emerging app issues has garnered positive feedback from developers, while DIVER aids in prioritizing issues through word clouds and line charts. The OPT-based approach enhances user-friendliness by providing customized instructions for review analysis through dynamic prompt generation.

Figure 10 Comparison of feature extraction tools.

Despite their progress in extracting features and requirements from app reviews, these tools face several limitations and challenges. These include the need for manual intervention or labeled data, difficulties with complex linguistic structures and context-dependent information, scalability issues, limitations in evaluation methods, interpretability challenges, generalizability concerns, and effectively incorporating user feedback and domain knowledge. Addressing these limitations is essential to enhance the practical application of these techniques in software requirements analysis and related fields. Many tools rely on manual intervention or labeled data for training and evaluation. For instance, the KEFE approach requires manually reviewing and labeling key features in app descriptions, which is time-consuming and subjective. Similarly, the RE-BERT method and other supervised learning techniques need high-quality annotated datasets, which can be challenging and resource-intensive in rapidly evolving or niche areas. Additionally, some methods struggle with complex linguistic structures, context-dependent information, or domain-specific terms. For example, the SAFE approach primarily uses part-of-speech patterns and predefined rules, potentially missing the nuances of natural language in user reviews. This leads to inaccurate feature or requirement extraction in scenarios with ambiguous, colloquial, or domain-specific language. Specific tools face scalability issues when handling large volumes of user reviews or real-time data streams. While tools like SOLAR and RISING have effectively managed extensive datasets, other methods may struggle with massive data volumes or real-time processing requirements.

The evaluation methodologies and metrics used to assess these techniques may not capture all aspects of their effectiveness or practical applicability. Standard metrics like precision, recall, and F1-score may not fully reflect the usefulness or relevance of the extracted features or requirements in real-world scenarios. Additionally, comparative evaluations against baseline approaches or state-of-the-art techniques may not comprehensively understand a technique’s limitations or challenges in specific contexts or application domains.

Some methods face interpretability or transparency challenges, particularly those relying on complex machine-learning models or black-box approaches. This lack of interpretability can hinder developers’ or analysts’ ability to understand the reasoning behind the extracted features or requirements, limiting trust in the results and hindering the adoption of these techniques in critical decision-making processes.

The generalizability of specific techniques across different domains, platforms, or languages may be limited. Techniques tailored for particular domains or languages may not perform well with user reviews from other domains or in different languages, necessitating additional adaptation. Moreover, incorporating user feedback and domain knowledge during feature extraction remains challenging. While some methods, like SAFER and RISING, integrate domain-specific constraints or semi-supervised learning, further exploration and improvement are needed to combine human expertise and contextual information effectively.

RQ5: What future research directions could address current gaps in capabilities for efficient and precise analysis of app reviews for requirements?

Emerging research directions aim to enhance app review analysis for requirements (RQ5). Future studies should focus on developing resilient and scalable methods to manage diverse and evolving user data while integrating human insights and domain knowledge. As mobile applications expand, analysis techniques must adapt to new domains, platforms, and languages without extensive retraining or manual effort, ensuring effective and accurate user requirement capture across various apps.

Improving the interpretability and transparency of feature extraction models is essential for fostering trust and facilitating adoption in software development. Developers and stakeholders must understand how requirements are extracted from user reviews and how they inform decision-making. Explainable AI techniques, such as model visualization and feature importance ranking, can make these models more accessible to non-experts (Chatzimparmpas et al., 2020).

Human-in-the-loop approaches could enhance model interpretability and transparency. By involving domain experts in the analysis process, researchers can ensure that the extracted requirements are meaningful and relevant to software development. These approaches can also help identify and correct model or underlying data biases (Mosqueira-Rey et al., 2023).

Additionally, research should focus on developing comprehensive evaluation frameworks that assess feature extraction’s accuracy and practical relevance. Techniques must be technically accurate and provide actionable insights that improve software development quality. Evaluation frameworks should include relevance, usefulness, and impact metrics to ensure the extracted requirements are valuable to developers and stakeholders.

Overall, future research in app review analysis should bridge the gap between technical accuracy and practical relevance. By developing robust, scalable, interpretable, and relevant techniques, researchers can address current gaps and drive the adoption of user review analysis in real-world settings.

Discussion and future research

The mapping study revealed diverse feature extraction techniques for analyzing mobile app reviews and eliciting software requirements, including topic modeling, collocation finding, association rule mining, aspect-based sentiment analysis, frequency-based methods, word vector-based approaches, and hybrids. Topic modeling, especially LDA, is prevalent for identifying latent topics in extensive user reviews but struggles with short or noisy texts and requires significant manual effort.

Collocation-finding techniques identify frequently co-occurring word patterns, effectively capturing domain-specific terminology or multi-word expressions. However, they may produce meaningless co-occurrences, requiring additional filtering. Association rule mining, though less explored, provides insights into interconnections between user experiences and requirements but faces challenges in handling noise and extracting precise relationships.

Aspect-based sentiment analysis captures nuanced, feature-specific feedback and sentiments, enabling targeted improvements and requirement prioritization, though it struggles with context and domain-specific terminology. Simple approaches like TF-IDF and POS tagging provide a baseline for feature extraction but may overlook semantic relationships and infrequent aspects.

Word vector-based techniques, such as word embeddings and language models, capture semantic and contextual information in user feedback, effectively analyzing sentiments and extracting key features. However, their effectiveness depends on training data quality and capturing domain-specific nuances. Hybrid approaches, combining multiple methodologies, leverage complementary strengths, offering more robust feature extraction and analysis solutions.

Regarding performance, tools like SIRA, SOLAR, MERIT, DIVER, SAFER, AOBTM, and IETI have outperformed others across various metrics such as precision, recall, and F1-score. Scalability is crucial, with tools like UisMiner and KEFE efficiently handling large volumes of reviews and others like SIRA, SOLAR, and the OPT-based approach demonstrating scalability with popular app data or few-shot learning.

Despite progress, challenges remain. Many techniques rely on manual intervention or labeled data, which is resource-intensive and subjective. Some approaches find it challenging to handle complex linguistic structures, context-dependent information, and domain-specific terminology. Evaluation methodologies may not fully capture practical applicability, and interpretability and transparency are concerns, especially for complex or black-box models. Generalizability across different domains, platforms, or languages is limited, and incorporating user feedback and domain knowledge remains challenging.

Future research should focus on creating robust, scalable methods to improve model interpretability and transparency for dynamic user reviews, integrating human insights and domain knowledge, and adapting to new domains, platforms, or languages without extensive retraining. Improving model interpretability and transparency is crucial for fostering trust and adoption in real-world software development. Explainable AI techniques and human-in-the-loop approaches could enhance model transparency. Additionally, developing comprehensive evaluation frameworks that capture both the accuracy and practical relevance of extracted features is essential.

The study highlights significant progress in feature extraction techniques for mobile app review analysis and software requirements elicitation. However, it underscores the need for continued research to address remaining challenges, focusing on developing robust, scalable, interpretable, and domain-adaptable techniques that effectively leverage user feedback and domain knowledge.

Threats to the validity

Recognizing potential validity threats is essential to maintain the integrity and strength of the conclusions drawn from our mapping study. By openly addressing these threats and detailing the steps taken to counteract them, we strive to strengthen the study’s dependability and applicability. This article identifies three primary threats to validity: Firstly, there is a possibility of missing relevant studies despite the comprehensive search strategy employed regarding the literature search and study selection process. To mitigate this threat, the search was conducted across multiple reputable digital libraries and databases, and the search strings were carefully constructed and iteratively refined. Additionally, backward and forward snowballing techniques were employed to identify potentially relevant studies that may have been overlooked in the initial database searches. Nevertheless, some studies may have been inadvertently excluded due to limitations in the digital libraries’ search terms or indexing mechanisms.

Secondly, the study selection process involved applying predefined inclusion and exclusion criteria, which may have introduced subjectivity and bias. The selection criteria were clearly defined to address this threat, and multiple researchers conducted the screening process independently. Discussions and consensus-building among the research team members resolved any disagreements or ambiguities.

Thirdly, the data extraction and synthesis process relied on accurately interpreting and representing the primary studies’ content. A standardized data extraction form was employed to mitigate potential threats to validity in this aspect, and multiple researchers carried out the data extraction process to ensure consistency and minimize individual biases. Moreover, regular team discussions and cross-checking mechanisms were implemented to resolve discrepancies or ambiguities in the extracted data.

The quality assessment of the primary studies was based on predefined criteria adapted from well-established guidelines in software engineering literature reviews. While these criteria aimed to capture relevant aspects of study quality, it is essential to acknowledge that some factors may have been overlooked. To mitigate this limitation, the quality assessment process involved multiple independent researchers to reduce individual biases. Despite the measures taken to address potential threats to validity, inherent limitations and biases may still exist. Therefore, the findings and conclusions should be interpreted within the specified scope and research questions, exercising appropriate caution.

In summary, while employing a rigorous and systematic approach, potential threats to validity were identified and addressed through carefully constructed search strategies, well-defined inclusion and exclusion criteria, standardized data extraction processes, quality assessment criteria, and consensus-building mechanisms among the research team. By acknowledging and mitigating these threats, the study aims to provide a comprehensive and reliable synthesis of the current state of feature extraction techniques and tools for mobile app review analysis while identifying opportunities for future research and improvements.

Conclusion

This study thoroughly evaluates automated and semi-automated methods for extracting features and software requirements from mobile app reviews. It groups these techniques into seven primary categories: topic modeling, collocation finding, association rule-based, aspect-based sentiment analysis, frequency-based, word vector-based, and hybrid techniques. The research identifies and discusses 48 tools and approaches that facilitate the implementation of these feature extraction methods. Various tools, such as SAFE, IDEA, AR-MINER, KEFE, CASPAR, ReUS, GuMa, SAFER, CLAP, CHANGEADVOSOR, MERIT, DIVER, SAFER (again), SIRA, and AOBTM, employ natural language processing, machine learning, and information retrieval techniques to analyze user reviews, extracting features, requirements, and feedback. These tools are assessed based on precision, recall, and F1-score for their performance, scalability, and user-friendliness. Some tools, including MERIT, DIVER, SAFER, SIRA, and AOBTM, have performed better than baseline tools like IDEA and SAFE in identifying emerging issues, recommending features, and extracting relevant information from app reviews. The study outlines the constraints and difficulties of current feature extraction methods from mobile app reviews. These limitations involve manual intervention or labeled data dependence, complex linguistics and contextual data handling issues, scalability, evaluation method limitations, interpretability challenges, generalizability concerns, efficient user feedback, and domain knowledge integration. In summary, the mapping study thoroughly explains the current state-of-the-art in mobile app review feature extraction, encompassing its strengths, weaknesses, and future research prospects.

Supplemental Information

Supplemental Information 1 Quality Assessment Scores of Selected Primary Studies.

Supplemental Information 2 Studies chosen for the mapping study.

Additional Information and Declarations

Competing Interests

Author Contributions

Data Availability

The authors declare that they have no competing interests.

Rhodes Massenon conceived and designed the experiments, performed the experiments, analyzed the data, performed the computation work, prepared figures and/or tables, authored or reviewed drafts of the article, and approved the final draft.

Ishaya Gambo conceived and designed the experiments, performed the experiments, analyzed the data, performed the computation work, prepared figures and/or tables, authored or reviewed drafts of the article, and approved the final draft.

Roseline Oluwaseun Ogundokun performed the experiments, analyzed the data, prepared figures and/or tables, authored or reviewed drafts of the article, and approved the final draft.

Ezekiel Adebayo Ogundepo performed the experiments, analyzed the data, prepared figures and/or tables, authored or reviewed drafts of the article, and approved the final draft.

Sweta Srivastava performed the experiments, analyzed the data, prepared figures and/or tables, authored or reviewed drafts of the article, and approved the final draft.

Saurabh Agarwal performed the experiments, analyzed the data, prepared figures and/or tables, authored or reviewed drafts of the article, and approved the final draft.

Wooguil Pak performed the experiments, analyzed the data, prepared figures and/or tables, authored or reviewed drafts of the article, and approved the final draft.

The following information was supplied regarding data availability:

This is a literature review.

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
