# Peer review of "Mobile app review analysis for crowdsourcing of software requirements: a mapping study of automated and semi-automated tools"

_PeerJ Computer Science, doi:10.7717/peerj-cs.2401_

## Round 0.1 · original submission · Minor Revisions

Please go through the reviewers' comments and addressed them as appropriate.
I suggest that the tables and figures in the paper should be revisited again to make sure they are readable and referenced in the paper at the right place.

Thanks for your hard work

Reviewer 1 ·

Basic reporting

Enhancing the related work section by incorporating discussions of more pertinent studies can provide a richer context and deeper insight into the existing literature.

Introduce a comprehensive summary through a diagram or a brief paragraph at the beginning of the document to outline the paper's structure. This strategy will significantly enhance the manuscript's readability, which is crucial due to its extensive length.

The text in Figures 5, 6, and 7 is currently illegible; enhancing the clarity and size of the text in these figures is essential for reader comprehension.

It is essential to thoroughly address and correct grammatical errors to ensure the manuscript is presented polished and professionally.

Experimental design

Enhance the flow and coherence of the manuscript by improving the transitions and linkages between paragraphs throughout the text.

Clarify the rationale behind conducting this review by emphasizing the necessity and significance of this study within the current research landscape.

Validity of the findings

The effectiveness of this review could be substantially improved by integrating additional visual aids that clearly demonstrate how different tools manage and interpret complex data sets.

·

Basic reporting

The review titled "Mobile App Review Analysis for Crowdsourcing of Software Requirements: A Mapping Study of Automated and Semi-automated Tools" covers a topic of broad and cross-disciplinary interest to researchers and practitioners in software engineering, requirements engineering, natural language processing, and data mining. It falls well within the scope of a multidisciplinary journal focused on computer science, software engineering, and related domains.

While previous literature reviews have explored app review analysis techniques, opinion mining in software development, user review classification, and sentiment analysis, this review identifies and addresses a specific gap. It provides a comprehensive evaluation and systematic comparison of the accuracy and performance metrics of various automated feature extraction methods tailored for deriving software requirements from mobile app reviews. This aspect has not been thoroughly examined in prior reviews, despite the recognized benefits of app reviews for requirements engineering.

The Introduction section effectively sets the context by establishing the importance of requirement engineering and the role of user reviews in identifying software features and requirements. It provides a comprehensive overview of existing literature while clearly identifying the research gap in the lack of a comparative assessment of fine-grained feature extraction methods from app reviews. The motivation for the study, to conduct an in-depth, side-by-side technique comparison and identify opportunities, best practices, and research gaps, is clearly articulated. The target audience, including researchers and practitioners in software engineering, requirements engineering, and related fields, is implicitly defined based on the subject matter and stated objectives.

Improvements for basic reporting: 1. Novelty and contribution: - Strengthen the novelty and contribution by providing a more detailed discussion on how this review differs from or complements previous related reviews, and highlight the new insights or perspectives it brings. 2. Introduction and motivation: - Provide a more explicit definition of the target audience. - Highlight the potential impact or implications of the work for the target audience. 3. Clarity and organization: - Conduct a detailed review of the writing style, logical flow, and structure of different sections to identify potential areas for improvement in clarity and readability. Overall, while the paper seems relevant and novel, potential improvements include emphasizing novelty, explicitly defining the target audience, highlighting implications, and ensuring clarity and organization throughout the manuscript.

Experimental design

1. The survey methodology appears consistent with a comprehensive, unbiased coverage of the subject. The authors have outlined a detailed and systematic mapping review process, including a rigorous literature search strategy, clear inclusion and exclusion criteria, and a well-defined data extraction and synthesis process. The literature search was conducted across multiple reputable digital libraries and databases, and the search strings were carefully constructed and iteratively refined. Additionally, backward and forward snowballing techniques were employed to identify potentially relevant studies that may have been overlooked in the initial database searches. The inclusion and exclusion criteria were explicitly defined and applied by multiple researchers to minimize bias. Overall, the methodology aims to ensure a thorough and unbiased coverage of the subject matter.

2. The sources are adequately cited throughout the paper. The authors have provided appropriate citations for the studies, techniques, and tools they discuss. Direct quotes from other sources are clearly marked and referenced, while paraphrased content is accompanied by relevant citations. The reference list at the end of the paper includes a comprehensive list of the cited works, adhering to proper citation practices.

3. The review is organized logically into coherent paragraphs and subsections. The paper follows a clear structure, starting with an introduction that provides background information and outlines the research questions and objectives. The methodology section meticulously describes the mapping review process, including the literature search strategy, inclusion and exclusion criteria, data extraction, and synthesis process. The results section is divided into subsections that systematically address each research question, presenting the findings in a organized manner. The discussion section provides an in-depth analysis of the results, highlighting strengths, limitations, and future research directions. The conclusion section summarizes the key insights and contributions of the study. Overall, the logical organization of the paper, with well-structured paragraphs and subsections, enhances the clarity and flow of the review.

Validity of the findings

1. The paper presents a well-developed and supported argument that meets the goals set out in the introduction. The authors systematically address each research question by conducting a comprehensive mapping study, providing a detailed analysis of identified feature extraction techniques, tools, and their performance in the context of mobile app review analysis for software requirements elicitation.

2. The discussion section highlights the significant strengths and limitations of the current techniques based on their methodology or evaluation results. The authors acknowledge remaining challenges, such as the need for manual intervention, difficulties with complex linguistic structures, scalability issues, interpretability challenges, and effectively incorporating user feedback and domain knowledge.

3. The conclusion section identifies unresolved questions, gaps, and future research directions. The authors outline several directions, including developing robust and scalable methods, improving model interpretability and transparency, developing comprehensive evaluation frameworks, and bridging the gap between technical accuracy and practical relevance. The paper emphasizes the need for continued research to address remaining challenges and develop techniques that are robust, scalable, interpretable, and domain-adaptable, while effectively leveraging user feedback and domain knowledge.

Validity of finding improvements
1. The paper presents a well-developed and supported argument that meets the goals outlined in the introduction. The methodology section outlines a rigorous and systematic approach, lending credibility to the findings. However, the authors could further strengthen the validity of the findings by providing more in-depth discussions on limitations and potential biases, including more quantitative comparisons and statistical analyses, and addressing potential threats to the validity of the mapping study itself.

---

## Round 0.2 · accepted · Accept

You have addressed all the comments. Thanks

Reviewer 1 ·

Basic reporting

no comment

Experimental design

no comment

Validity of the findings

no comment

Additional comments

no comment